# Full-length transcriptome reconstruction reveals a large diversity of RNA and protein isoforms in rat hippocampus

Xi Wang[1,2,7]*, Xintian You[1,3,7], Julian D. Langer[4], Jingyi Hou[1], Fiona Rupprecht[4], Irena Vlatkovic[4], Claudia Quedenau [1], Georgi Tushev [4], Irina Epstein[4], Bernhard Schaefke[5,6], Wei Sun[5], Liang Fang[5,6], Guipeng Li [5,6], Yuhui Hu[5], Erin M. Schuman [4] & Wei Chen [5,6]*

Gene annotation is a critical resource in genomics research. Many computational approaches have been developed to assemble transcriptomes based on high-throughput short-read sequencing, however, only with limited accuracy. Here, we combine next-generation and third-generation sequencing to reconstruct a full-length transcriptome in the rat hippocampus, which is further validated using independent 5′ and 3′-end profiling approaches. In total, we detect 28,268 full-length transcripts (FLTs), covering 6,380 RefSeq genes and 849 unannotated loci. Based on these FLTs, we discover co-occurring alternative RNA processing events. Integrating with polysome profiling and ribosome footprinting data, we predict isoform-specific translational status and reconstruct an open reading frame (ORF)-eome. Notably, a high proportion of the predicted ORFs are validated by mass spectrometry-based proteomics. Moreover, we identify isoforms with subcellular localization pattern in neurons. Collectively, our data advance our knowledge of RNA and protein isoform diversity in the rat brain and provide a rich resource for functional studies.

[1] Max Delbrück Center for Molecular Medicine, 13125 Berlin, Germany. [2] German Cancer Research Center, 69120 Heidelberg, Germany. [3] Max Planck Institute for Molecular Genetics, 14195 Berlin, Germany. [4] Max Planck Institute for Brain Research, 60438 Frankfurt, Germany. [5] Department of Biology, Southern University of Science and Technology, 518055 Shenzhen, Guangdong, China. [6] Medi-X Institute, SUSTech Academy for Advanced Interdisciplinary Studies, Southern University of Science and Technology, 518055 Shenzhen, Guangdong, China. [7]These authors contributed equally: Xi Wang, Xintian You. *email: xi.wang@dkfz.de; chenw@sustech.edu.cn

The structural annotation of genes serves as an important resource in the genomics era. Prior to the advent of next-generation sequencing (NGS) technologies, gene annotation proceeded relatively slow and largely relied on expressed sequence tags (ESTs)[1], protein sequences[2], in silico prediction[3], and their combinations[4]. Recently, NGS-based RNA-seq has been used to reconstruct transcriptomes by assembling sequencing reads with[5] or without[6] reference genomes, largely expanding the complexity of transcriptomes in human and other organisms[7–9]. However, transcriptome diversity owing to alternative transcription start sites, alternative splicing of exons, and/or the use of different poly(A) sites is often difficult to capture and characterize using NGS data, due to their relatively short read length (typically ≤ 400 nt)[10] in comparison to the length of mature transcripts (median > 2500 nt). Therefore, coupled alternative RNA processing events that occur over long distances cannot be simultaneously captured by single short reads[11]. In such cases, transcriptome reconstruction algorithms cannot provide unambiguous solutions[5]. In general, the quality of isoform assembly is inversely correlated with the transcriptome complexity[12,13].

Despite the technical challenge, there is an ever-increasing interest to investigate and determine transcripts in full-length, which allows for the identification of coupled alternative RNA processing events thereby discovering coordination between different regulatory processes, including coding capacity, RNA localization, translation and stability. For example, it has been reported that alternative splicing of internal/last exons is coupled with alternative polyadenylation sites (pAs) to alter simultaneously coding sequences and 3′-untranslated regions (UTR)[14,15]. Cis-elements residing in 3′ UTRs can affect RNA stability[16,17], translational efficiency[18–20], and subcellular localization[21–23]; therefore, such coupling could generate distinct protein isoforms with differential translational efficiency from RNAs with differential stability and/or subcellular localization[23–25]. In addition, the knowledge of full-length mRNA sequences or at least full-length ORFs is essential to understand the complete amino acid sequences of encoded proteins. Moreover, to what extent transcriptome diversity is translated into proteome diversity is still debated[26–28]. Clear, unbiased answers to the abovementioned questions require the knowledge of full-length transcriptomes.

The more recently developed third generation sequencing (TGS) technologies, including the Pacific Biosciences (PacBio)[29–35] and Oxford Nanopore[36] platforms, can generate longer reads, reaching up to tens or even hundreds of kilobases[10,37]. Such read length is sufficient to cover most RNA transcripts in full length. Although these technologies have recently been used to characterize full-length transcripts either in a targeted manner or on a genome-wide scale[29–36], challenges exist in two respects. First, the sequencing error rate of TGS is significantly higher than NGS, which compromises read alignment or assembly[31]. One way to increase the read accuracy is to sacrifice the advantage of read length. For instance, PacBio can generate high-quality circular consensus sequences (CCSs) by requiring the polymerase to run through the sequencing insert multiple times. However, if sequencing inserts are long, the polymerase may fail to go through the insert multiple times, and only subreads, generated by one-pass polymerase runs, are obtained. As a result, the CCS sequences could be significantly shorter than the subreads[32]. Alternatively, the same samples could be sequenced at the same time by, for example, Illumina technology and thereafter the high-quality Illumina reads could be used to correct the TGS sequencing errors (e.g., LSC[38], pacBioToCA[31], proovread[39], as well as others reviewed and compared in ref. [40]). Second, the TGS throughput is significantly lower than that of Illumina[41], which, if used on a genome-wide scale, would be difficult to capture transcripts of low abundance[30]. Moreover, what is often lacking in previous TGS transcriptome studies, is the orthogonal validation of transcript ends[29,32,42].

Although the rat has long been used as a model organism, its gene annotation is not comparable with that of human and mouse. This largely restricts its potential usage in many fields, including molecular neurobiology given the highest transcriptome complexity in neuronal tissues. To address this issue and above-stated technical challenges, we develop a hybrid-sequencing workflow for reconstructing full-length transcripts (FLTs) and apply it to annotate the FLTs expressed in rat hippocampus, which are then validated by orthogonal approaches. With the FLT collection, we discover co-occurrence of alternative RNA processing events. Furthermore, integrating with polysome profiling and ribosome footprinting data, we provide insightful observations on the isoform–specific translational status, as well as ORFeome diversity. Importantly, a large fraction of our newly identified ORFs are supported by mass spectrometry-based proteomics datasets. Finally, we identify isoforms with specific subcellular localization patterns in neurons. Altogether, our data expand the current rat gene annotation and provide a rich resource for future functional studies. Moreover, our pipeline can be easily adapted to uncover the complex transcriptome/ORFeome landscapes expressed in other tissues from other organisms.

## Results

**A hybrid-sequencing workflow for FLT reconstruction.** Aiming for sequencing RNA transcripts in full length, we developed a workflow that combines the unique technical strengths of Illumina and PacBio sequencing technologies, i.e., high sequencing accuracy and long read length, respectively. Our workflow has two main features (Fig. 1a): First, cDNA normalization and size fractionation to facilitate the profiling of diverse transcripts at an affordable throughput; second, a computational approach to correct PacBio sequencing errors by using Illumina reads.

We applied the workflow to adult rat hippocampal tissue. In total, we sequenced four PacBio libraries of different cDNA sizes (Supplementary Table 1). Incorporating the size fractionation helps to avoid amplification and sequencing bias. While shorter-than-expected subreads were observed to different extents in the four libraries, the overall subread length showed a tendency to fit the expected size (Supplementary Fig. 1a), and the reads were predominantly derived from genes of the expected length (Supplementary Fig. 1b). We speculated that those shorter-than-expected reads were largely due to polymerase drop-off during the PacBio sequencing; therefore these reads were not used for FLT annotation (see the next section).

The cDNA in the PacBio library was normalized using a duplex-specific nuclease[43] (see Methods section). As a result, the range of PacBio reads per gene spanned only three orders of magnitude, considerably less than the seven orders of magnitude of the gene expression dynamic range measured by Illumina sequencing of the non-normalized library (Supplementary Fig. 1c, d). As shown in Fig. 1b and Supplementary Fig. 1e, after the cDNA normalization, those highly expressed transcripts, which would otherwise consume many sequencing reads, were largely depleted. Hence, after normalization we obtained more reads from lowly expressed genes, enabling us to sample relatively rare transcripts.

Using Illumina reads to correct PacBio sequencing errors (see Methods section), we substantially improved the sequence accuracy (Fig. 1c, Supplementary Fig. 1f for an example). As a result, the percentage of alignable reads (Fig. 1d), alignment coverage (Supplementary Fig. 1g) and precision of read alignment at annotated splicing sites (Supplementary Fig. 1h) were

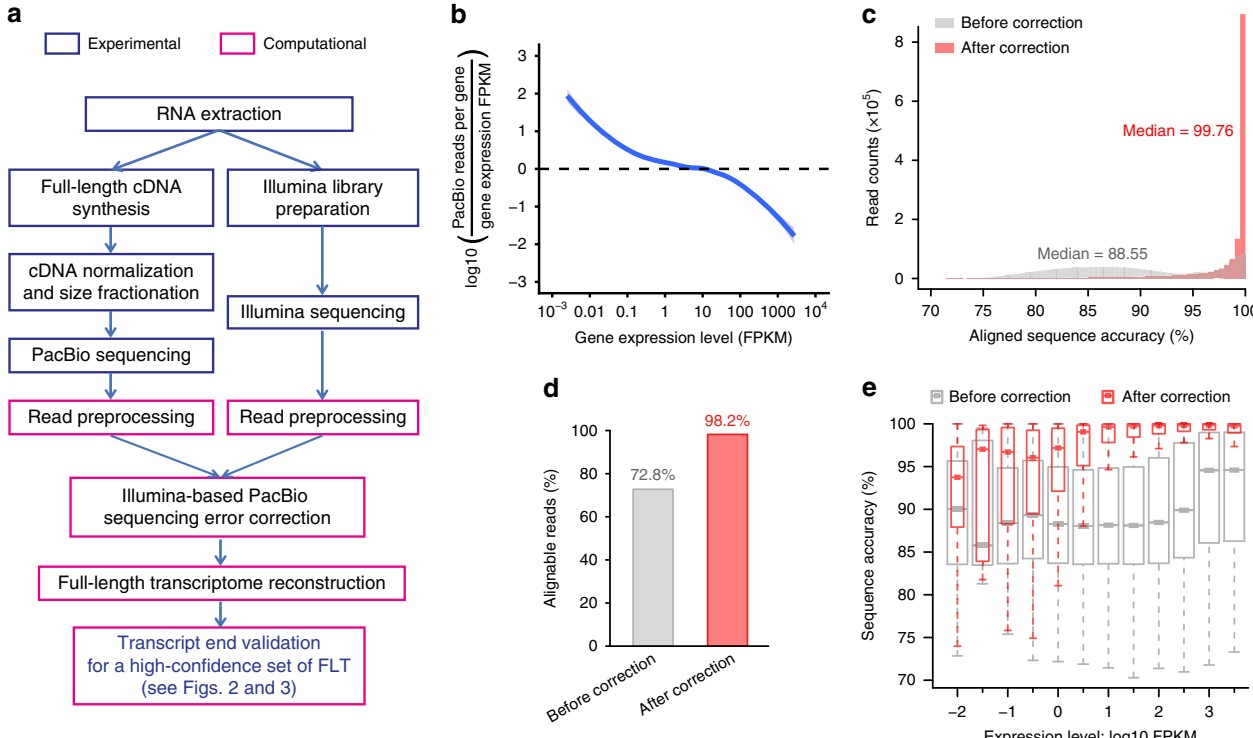

**Fig. 1** The hybrid sequencing workflow for FLT annotation. **a** The flow chart of the hybrid sequencing workflow, including the experimental procedures (blue rectangles) comprised of PacBio sequencing and Illumina sequencing, and the computational components (pink rectangles) for Illumina-based PacBio error correction, followed by full-length transcriptome reconstruction, transcript end validation, and isoform diversity characterization. **b** The ratio of PacBio read count to gene expression level (estimated based on Illumina sequencing data) was plotted against the gene expression level. Highly expressed transcripts were largely depleted in PacBio sequencing after the cDNA normalization. **c** Histograms compare PacBio read accuracy before and after the error correction. After the correction, the final sequence accuracy was increased from 88.55% to 99.76%. The colors filled in the histograms are half-transparent, so that dark red in the histograms indicates the overlay. **d** Bar plots showing the percentage of alignable reads before and after error correction (72.8% vs. 98.2%). **e** The increased sequence accuracy after error correction was plotted against gene expression levels. Box edges represent quartiles, whiskers represent extreme data points no more than 1.5 times the interquartile range. Source data for panel **b** are provided in a Source Data file

improved. As genes of higher expression levels obtained more Illumina reads for error correction, we observed a positive correlation between the improved accuracy and the expression levels, whereas the accuracy of the raw reads was independent of gene expression (Fig. 1e). Except for genes expressed at very high levels (FPKM > 100), the performance of error correction could be further improved with higher Illumina sequencing depth.

After error correction, we aligned the PacBio reads against the rat reference genome, identical alignments were collapsed, and reads aligned to the same loci of the same strand were clustered (see Methods section). Hereafter, we define collapsed reads as transcripts, and read clusters as gene loci. At this stage, we identified 102,377 transcripts from 22,629 gene loci.

**High-confidence FLTs improve rat RNA isoform annotation.** To discard truncated reads caused by incomplete reverse transcription or PacBio sequencing, we generated cap analysis gene expression (CAGE) data[19] (see Methods section) and used our previously published 3′end sequencing (3′-seq) data from the rat hippocampus[23] to characterize the transcript 5′-ends and 3′-ends, respectively (Fig. 2a). In total, the CAGE and 3′-seq data yielded 37,426 and 23,229 clusters, respectively. At the transcription start sites (TSSs), we found a substantial overlap between our PacBio transcript 5′-ends and CAGE clusters (Fig. 2b). Among the non-overlapping ones, PacBio transcript 5′-ends were more frequently located downstream of CAGE clusters, likely due to reverse transcriptase or polymerase drop-off. A similar phenomenon was observed when comparing to the annotated TSSs, though the

distance distribution was less sharp and there were fewer overlaps within a 50-nt distance (Fig. 2b). In comparison to the 5′-ends, better agreement was achieved at the transcript 3′-ends (Fig. 2c) and again, there were more overlaps between PacBio transcript 3′-ends and 3′-seq clusters than that between PacBio transcript 3′-ends and annotated transcription end sites (TESs). Collectively, these analyses showed the high concordance between PacBio transcript ends and CAGE/3′-seq clusters, and indicated the incompleteness of current rat TSS/TES annotation.

To determine high-confidence FLTs, we only retained transcripts whose 5′-ends and 3′-ends were both within 50-nt from CAGE and 3′-seq clusters. The final set contained 28,268 transcripts derived from 6,380 RefSeq gene loci and 849 unannotated loci, resulting in an average diversity of 3.91 isoforms per gene locus, a large increase in isoform diversity compared to rat RefSeq/Ensembl annotation (Fig. 3a; see also Supplementary Fig. 2a, b for comparison to mouse and human gene annotation, respectively). For instance, our FLT contained eight transcript isoforms of the gene *Nrsn1* (Neurensin-1), which started from two distinct TSSs validated by CAGE and ended at one TES confirmed by 3′-seq (Fig. 2d). In comparison, the RefSeq annotation contains only one isoform, and the annotated TES was not used in our sample. In *Rpl21* (large ribosomal subunit protein 21), we found one additional distal TES and a number of alternative first exons, whereas RefSeq contains only one annotated isoform (Fig. 2e). Finally, for the gene *Sirpa* (signal regulatory protein α), which encodes a transmembrane protein predominantly expressed in neurons and present in dendritic

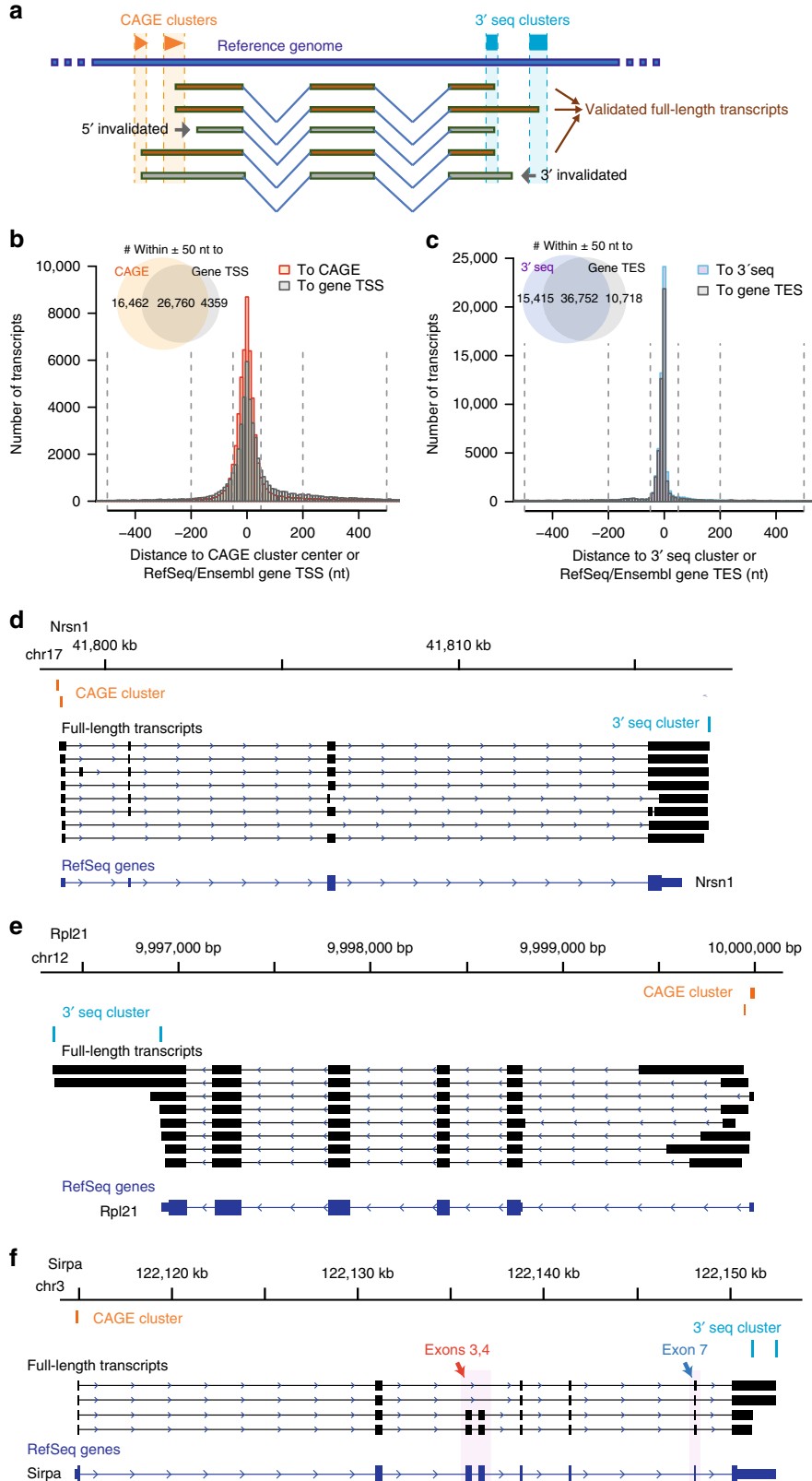

and/or axonal regions[44], we identified four isoforms with two distinct 3′-ends, all divergent from the one annotated in RefSeq. As shown in Fig. 2f, the inclusion of two internal exons 3 and 4 seems to be associated with a shortened 3′UTR, whereas the choice of alternative 5′splice sites of exon 7 was independent of the usage of the two alternative TESs.

We then checked whether any systematic biases were introduced into our FLT transcripts. First, concerning the size and position of exons and introns, we found no large differences between our FLT and RefSeq/Ensembl annotation (Supplementary Fig. 2c, d). Then, we evaluated the 3-divisibility of cassette exon length[45] and found a significant enrichment of 3-divisible

**Fig. 2** The high-confidence set of FLTs. **a** The scheme of full-length transcript end validation and filtering, based on CAGE and 3'seq data. In the example transcripts, the third and last transcripts from top to bottom are invalidated, due to their 5'-ends and 3'-ends are not supported by CAGE and 3'seq clusters, respectively. These transcript fragments were excluded in the downstream analysis. **b** Histograms show the genomic distance between PacBio transcript 5'ends and CAGE clusters, as well as the distance between PacBio transcript 5'ends and annotated gene TSSs. Positive distance indicates PacBio transcript 5'ends located downstream of CAGE clusters (or annotated TSSs) and negative distance indicates PacBio transcript 5'ends located upstream of CAGE clusters (or annotated TSSs). The inlet Venn diagram shows that there were more PacBio transcript 5´ends located within 50-nt from CAGE clusters than located within 50-nt from annotated TSSs. **c** Histograms show the genomic distance between PacBio transcript 3'ends and 3'seq clusters, as well as the distance between PacBio transcript 3'ends and annotated gene TESs. Positive distance indicates PacBio transcript 3'ends located downstream of 3'seq clusters (or annotated TESs) and negative distance indicates PacBio transcript 3'ends located upstream of 3'seq clusters (or annotated TESs). The inlet Venn diagram shows that there were more PacBio transcript 3'ends located within 50-nt from 3'seq clusters than located within 50-nt from annotated TESs. **d–f** Three example genes of the high-confidence FLT. Tracks from top to bottom include genomic coordinates, CAGE clusters, 3'seq clusters, full-length transcripts, and RefSeq genes. The three genes are **d** *Nrsn1* (Neurensin-1), a gene specifically expressed in neurons, **e** *Rpl21*, the large ribosomal subunit protein 21, and **f** *Sirpa*, signal regulatory protein α. Source data for panels **b** and **c** are provided in a Source Data file

cassette exons in the coding region (Supplementary Fig. 2e), suggesting that such cassette exons were selected to maintain ORF integrity. In particular, cassette microexons of 3–27 nt in length had a higher probability (98 out of 141) to be 3-divisible, consistent with the observations by Irimia et al.[46]. Moreover, we found a comparable conservation pattern at splicing sites, cassette exons, retained introns, as well as tandem 5´UTRs and tandem 3′ UTRs between our FLT and RefSeq/Ensembl annotation (Supplementary Fig. 2f–h). Finally, for both our FLT collection and Ensembl annotation in mouse/human, the isoform diversity in a gene was positively correlated with its maximum transcript length and the maximum exon number (Supplementary Fig. 2i), suggesting the observed diversity could be due, at least in part, to an enhanced opportunity for a transcript to undergo alternative RNA processing. Taken together, these analyses suggest no obvious bias imposed by our FLT reconstruction process.

**Co-occurrence of alternative RNA processing events**. We compared every FLT to the RefSeq transcript with the highest similarity. Unexpectedly, of those derived from RefSeq gene loci, 93% deviated from the annotated transcripts by at least one alternative RNA processing event (Fig. 3b, Supplementary Fig. 3a). Deviation by multiple alternative events accounted for 61% of these transcript variants, with approximately 5% exhibiting greater than five alternative events. For these, accurate assembly would be difficult, if not entirely impossible, to achieve with conventional NGS methods.

Based on their alternative processing events, we grouped the FLT variants into distinct categories (Fig. 3c, Supplementary Fig. 3b) and investigated the enrichment or depletion of the co-occurrence of all event pairs in those isoforms consisting of multiple alternative events. Interestingly, we observed a significant co-occurrence between alternative 5'-ends and 3'-ends, whereas the alternative usage of transcript ends did not co-occur with internal splicing events (Fig. 3d, Supplementary Fig. 3c). Among internal splicing events, the co-occurrence happened only between alternative 5' and 3' splice sites, and between multiple intron retention events (Fig. 3d, Supplementary Fig. 3c). Taking the most abundant isoforms in our FLTs as the reference for analyzing alternative processing events, similar results were observed (Supplementary Fig. 3f).

To further study the coordinated usage of transcript ends and alternative splice sites, we split alternative first/last exons and tandem 5'/3'UTRs into distal and proximal groups, and alternative splice sites into upstream and downstream sites (Supplementary Fig. 3d, e). For alternative TSS and TES, we observed a small but significant co-occurrence between proximal tandem 5'UTR and proximal tandem 3'UTR, and between distal alternative first exons and distal tandem 3'UTR (Supplementary Fig. 3d). Similarly, upstream/downstream alternative donor sites

tended to co-occur with downstream/upstream alternative acceptor sites (Supplementary Fig. 3e), which suggested that the observed co-occurrence between alternative splicing sites could not simply be due to ambiguous splice junction assignments.

**Diverse isoform-specific translational status**. To address the impact of the isoform diversity on translation, we performed polysome profiling (polysome-seq), and deeply re-sequenced previously generated ribosome footprinting (ribo-seq) libraries from rat brains[47] (see Methods section). Computationally, we first predict isoforms with active translational status and then reconstruct an ORFeome to investigate its diversity (Fig. 4a). The purpose of the prediction of isoforms with translationally active status was twofold: to dissect alternative events associated with active translation, and to prioritize a set of transcripts for ORF prediction.

To predict the translational status, we built supervised machine learning models based on the polysome-seq data. The classification models were trained by protein coding and noncoding isoforms annotated in Ensembl with a cross-validation accuracy of $86.31 \pm 0.33\%$ (see Methods section). Of the 28,268 FLTs, 3712 were excluded due to low expression level (Fig. 4b). For the remaining 24,556 transcripts, the classifiers predicted 16,976 with translationally active status, whereas the remaining 7580 (30.9%) were translationally inactive (Fig. 4b). At gene level, among 7229 loci, 6670 had at least one translationally active isoform.

Comparing the type of alternative processing events between isoforms with translationally active or inactive status, we found that intron retention, exon skipping, and alternative 5'/3' splicing sites were depleted in the active transcripts (Fig. 4c), largely because these events could likely disrupt the canonical ORFs by introducing frameshifts (e.g., Supplementary Fig. 2e). Indeed, the percentage of 3-divisible cassette exons in the CDS was significantly higher in the active isoforms than in the inactive ones (49.5% vs. 40.2%, $p$ value = 0.039, Fisher's exact test). Moreover, the retained introns in the active transcripts were significantly shorter than in the inactive ones (median = 102 vs. 129, $p$ value = 7e−9, Mann–Whitney $U$ test) and more biased towards either 3' or 5' end to avoid occurrence in CDS regions ($p$ value = 5e−12, Kolmogorov–Smirnov test). In contrast, tandem 5'/3' UTRs, which did not affect CDS, were enriched in translationally active transcripts (Fig. 4c). In addition, alternative first exons, which could either affect CDS or not, were distributed similarly between the active and the inactive isoforms (Fig. 4c).

**Discovery of ORF variants and novel ORFs**. Previous studies have shown that computational algorithms such as the ORFscore are able to detect translating ORFs by leveraging the 3-nt periodicity of Ribo-seq footprints[48]. Based on our data, we first

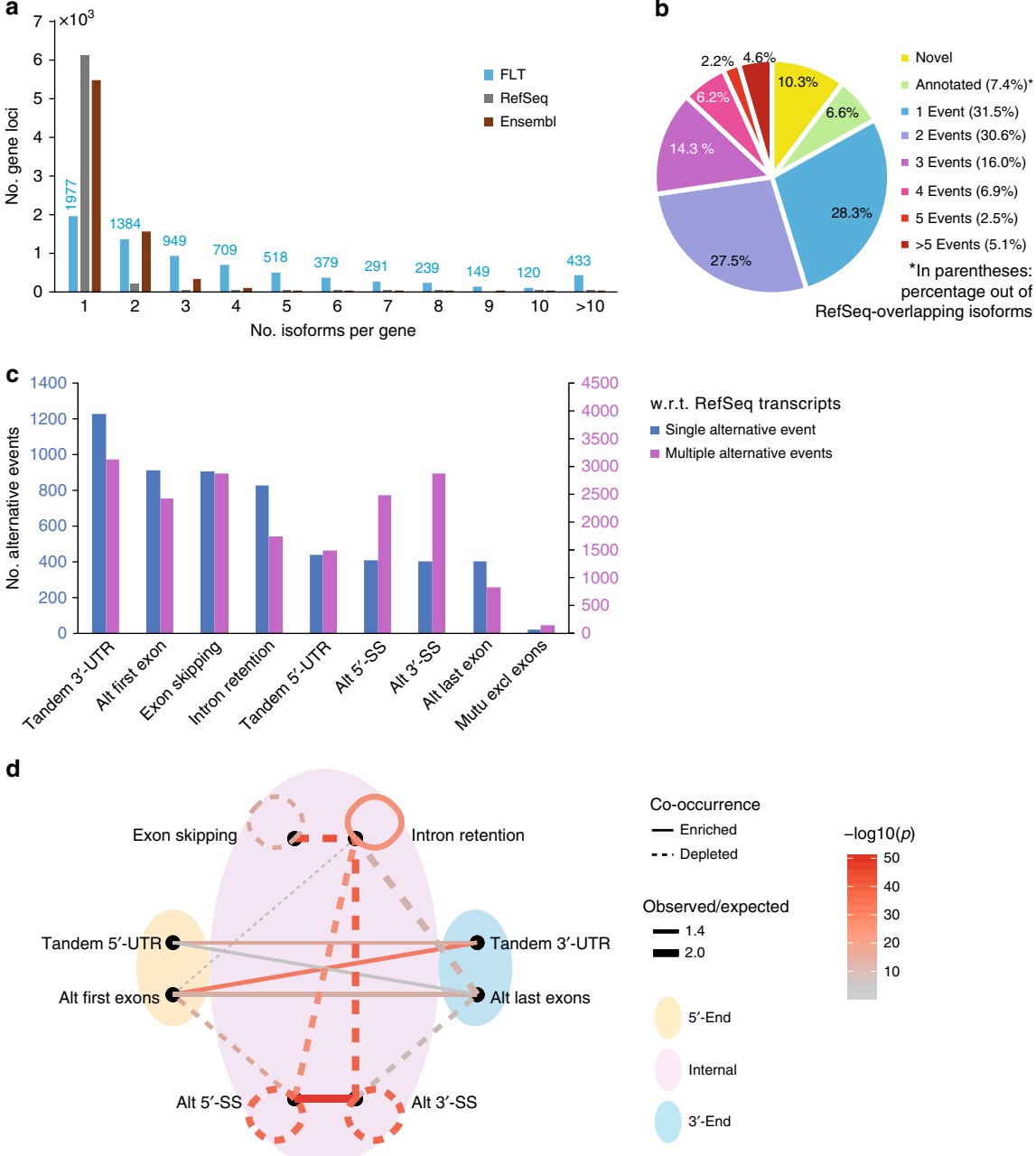

**Fig. 3** RNA isoform diversity and co-occurrence of alternative RNA processing events. **a** Number of isoforms per gene in our FLT, compared to rat RefSeq and Ensembl annotation. See also Supplementary Fig. 2a, b for the comparisons to mouse and human RefSeq/Ensembl annotation. **b** Pie chart shows the proportion of our FLT transcripts categorized with the respect to RefSeq curated isoforms. The categories include novel loci (non-overlapped with annotated RefSeq loci), annotated isoforms, and isoform variants that deviated from their closest RefSeq isoforms by 1, 2, 3, 4, 5, and >5 alternative RNA processing events. Shown in the pie chart are percentages out of all the FLT isoforms, and shown in the parenthesis are percentages out of RefSeq-overlapping isoforms. **c** Histograms show the number of transcript variants harboring different types of alternative events, including, from left to right, tandem 3′UTR, alternative first exons, exon skipping, intron retention, tandem 5′UTR, alternative 5′ splice sites, alternative 3′ splice sites, alternative last exons, and mutually exclusive exons. In the histograms, transcript variants were sorted into isoforms deviated from their closest RefSeq transcripts by single alternative event (blue) and multiple events (purple). Shown were only isoforms with TPM > 1; see Supplementary Fig. 3b for all the isoforms. **d** Network diagram shows the co-occurrence of RNA processing alternative events. Line type: enrichment or depletion of the co-occurrence; line width: the ratio between observed co-occurrence and expected co-occurrence; line color: −log10(*P* values) of the enrichment or depletion (one-tailed binomial tests). Shown were only isoforms with TPM > 1; see Supplementary Fig. 3c for all the isoforms. Source data for panels **a** and **d** are provided in a Source Data file

verified that ORFscores indeed had strong discriminative power to separate ORFs used for translation (ORFscore > 5; Fig. 5a right), ORFs in untranslated regions (−6 < ORFscore < 5; Fig. 5a middle), and ORFs overlapping with translating ORFs but in different frames (ORFscore < −6; Fig. 5a left). In addition, ORF

length has also been shown to be an important factor in coding potential prediction[49,50]. Here as often used in previous studies, if there were multiple ORFs in one transcript with ORFscore > 5, the longest ORFs were kept. Indeed, the longest ORFs showed a very similar distribution of ORFscores compared to the ORFs

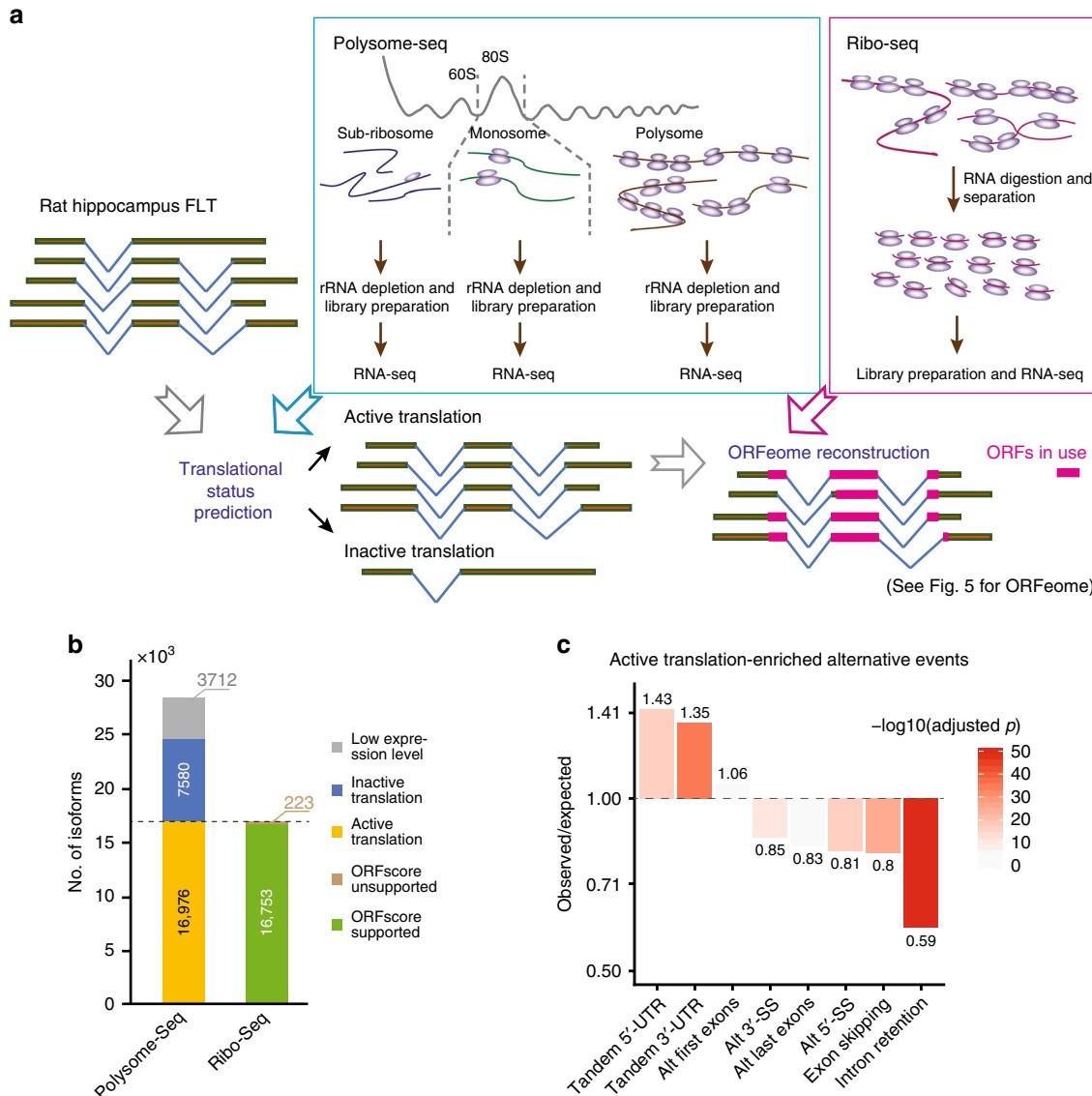

**Fig. 4** Diverse isoform-specific translational status. **a** Scheme of the integrative analysis of our FLT, polysome-seq and ribo-seq, in order to identify isoform-specific translational status and then reconstruct the ORFeome. **b** Bar plots show the proportion of our FLT isoforms of different translational status, predicted based on polysome-seq, and the proportion of predicted translationally active isoforms supporting by ribo-seq. **c** Bar plots show fold-changes between observed and expected alternative RNA processing events occurred in translationally active isoforms. Colors show the –log10 transformed Benjamini-Hochberg adjusted P values (Fisher's exact tests). Source data for panel **c** are provided in a Source Data file

annotated in RefSeq (Supplementary Fig. 4a, b). With the ORF-score threshold at 5, we detected translating ORFs in nearly all the translationally active isoforms predicted by polysome-seq data (16,753 out of 16,976, 98.7%; Fig. 4b). In the translationally inactive isoforms we found that the longest ORFs were significantly shorter and had significantly lower ORFscores ($p < 2.2 \times 10^{-16}$, Mann–Whitney $U$ test; Supplementary Fig. 5a, b), demonstrating that the polysome-seq prediction is well supported by the independent ribo-seq data.

Next, we asked whether the translationally active transcripts might be selected via coordinated RNA processing events to protect their ORFs. For this purpose, we generated in silico all possible isoforms by random selection of alternative exon blocks to the scaffold comprising all constitutive exons. Comparing the isoforms in the translationally active pool to all random isoforms, we found that the translationally active isoform always achieved higher ORFscores and longer ORF lengths ($p < 2.2 \times 10^{-16}$, Mann–Whitney $U$ test; Supplementary Fig. 5c, d), suggesting

that the capability to produce genuine proteins is enhanced by proper RNA processing.

We then reconstructed the ORFeome. Since different RNA isoforms may harbor the same ORF, we collapsed all putative ORFs to form a unique set consisting of 13,093 ORFs from 6,560 gene loci, resulting in an average of 1.99 ORFs per gene. Compared to 3.91 transcript isoforms per gene locus, the diversity in ORF was almost halved. Nonetheless, the ORF diversity was significantly higher than that in RefSeq (1.05) and Ensembl (1.34). As shown in Figs. 5b, 60.1% genes in our FLT dataset harbored more than one ORF, whereas only 3.6% and 25.8% genes in RefSeq and Ensembl exhibited multiple ORFs, respectively.

To further investigate how the putative ORFs varied from the annotated ones, we defined various types of ORF variants and novel ORFs, as shown in Fig. 5c. Interestingly, less than one third of the putative ORFs were annotated in RefSeq (29.4%), whereas the majority were ORF variants (61.1%) and about one tenth were

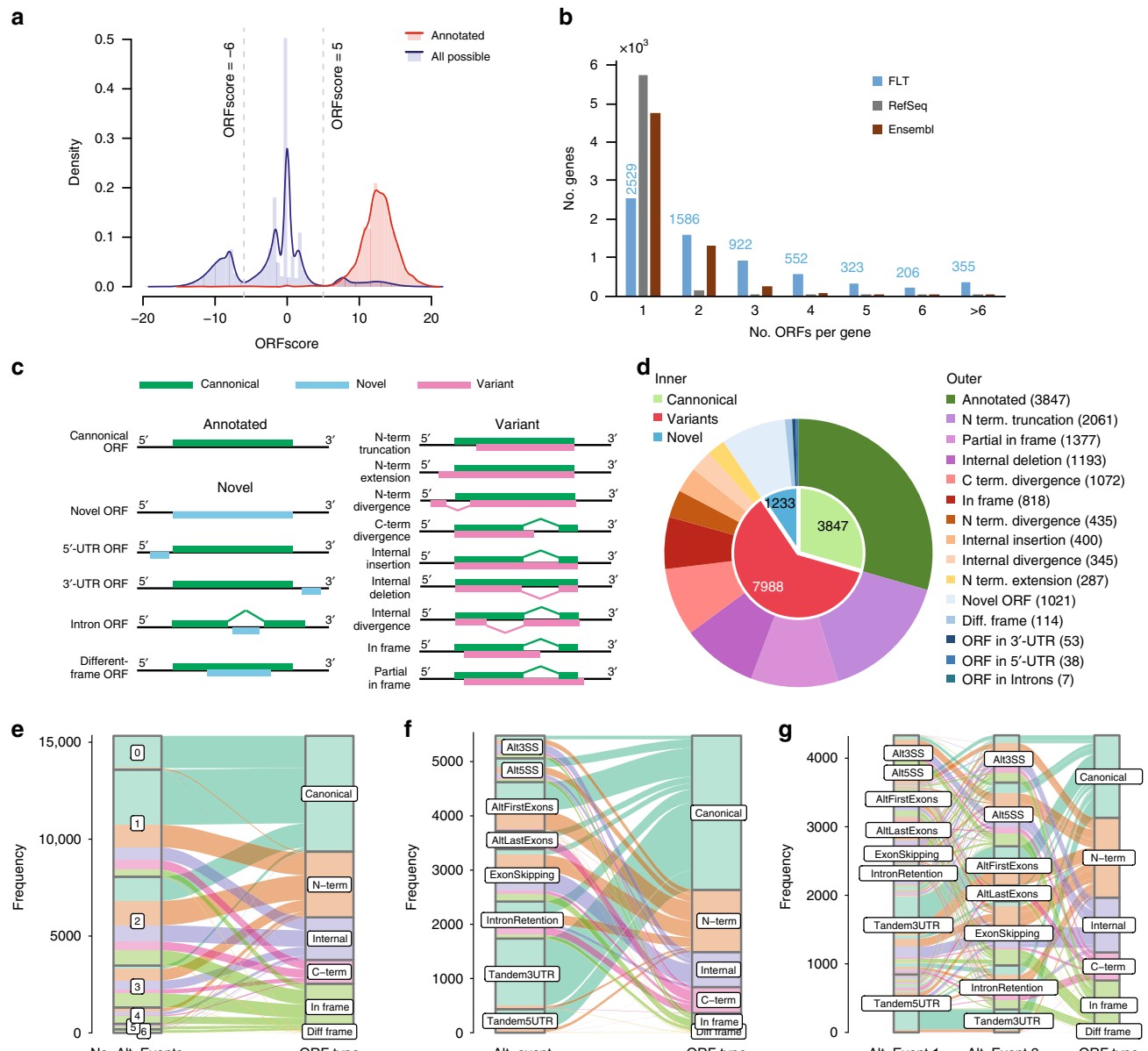

**Fig. 5** ORFeome reconstruction identifies unannotated ORF variants and novel ORFs. **a** Histograms show the distribution of ORFscores of annotated ORFs (red) and all possible ORFs in our FLT (blue). The distribution demonstrates the discriminative power of ORFscore at 5 to classify whether ORFs were translated or not. **b** Number of ORFs per gene in our ORFeome, compared to that in RefSeq/Ensembl annotation. **c** Scheme of ORF types, including annotated ORFs, ORF variants, and novel ORFs. **d** Pie charts show the proportion of different ORF types in our ORFeome. **e–g** Alluvial diagrams show the impact of RNA isoform diversity on the diversity of ORF variants, including **e** the impact of the number of alternative events, **f** the impact of single alternative events, and **g** the impact of pairs of alternative events. ORF types: Canonical, annotated ORF; N-term, including N-terminal truncation, N-terminal extension, and N-terminal divergence; Internal, including internal insertion, internal deletion, and internal divergence; C-term, C-terminal divergence; In frame, including in frame and partial in frame; Diff frame, different frame. Source data for panel **b** are provided in a Source Data file

novel (9.5%) (Fig. 5d). Of note, the percentage of our ORF collection being annotated in RefSeq (29.4%) is dramatically higher than that of our FLT collection (7%). This is because many alternative RNA processing events, such as tandem UTRs, do not affect ORFs (Fig. 5e–g).

Among the nine sub-categories of ORF variants, approximately one quarter (25.8%) were N-terminal truncations (Fig. 5d), where a start codon downstream of the annotated one was used. This phenomenon is in concordance with our previous observation in NIH-3T3 cells, where translation initiation site (TIS) profiling detected a large number of TISs downstream of annotated ones[19]. Mechanistically, this could be caused by using downstream alternative first exons (Fig. 5f). In addition to N-terminal

truncated proteins, divergence from and extension to annotated N-termini were also observed (5.45% and 3.59%, respectively). These N-terminal variants were mainly caused by one or two alterative events (Fig. 5e), including alternative first exons, and exon skipping/intron retention which disturbed the sequences around the annotated TISs (Fig. 5f, g). We speculate that many protein isoforms with different N-termini might alter protein localization due to the presence/absence of signal peptide (Supplementary Data 1). The next categories of ORF variants lay in internal parts, comprising internal deletion (14.9%), insertion (5.0%), and divergence (4.3%). These were predominantly caused by inclusion/exclusion of cassette exons (Fig. 5f). Moreover, we found 13.4% of ORF variants harbored different C-

termini from annotated ones, largely caused by alternative last exons or intron retentions near the stop codon (Fig. 5f). Finally, complex combinations of different types of ORF variants also existed in the dataset; these could be further classified into partial in-frame (17.2%) and in-frame shifts (10.2%). Often, these ORF variants lay in transcripts differing by at least two alternative events (Fig. 5e, g).

Of the novel ORFs, the majority were ORFs derived from novel gene loci (81.1%). The second sub-category was those overlapping with annotated ORFs but in different frames (9.1%; Fig. 5d). Finally, we found ORFs originating from regions annotated as un-translated (4.3% from 3′UTRs, 3.1% from 5′UTRs and 0.6% from introns), which tended to have short length and lower ORFscores (Supplementary Fig. 6) but might produce smaller peptides of functions[48,51]. Moreover, they also showed in general lower PhyloCSF[52] conservation scores (Supplementary Fig. 6e), with a few exceptions indicating potentially conserved functions (Supplementary Data 2)[51,53].

Furthermore, we re-analyzed our previously generated mass spectrometry (MS)-based proteomics datasets from rat primary hippocampal neurons[54] for peptides that support the reconstructed ORFeome (see Methods section; Supplementary Data 3). In order to avoid annotated ORFs confounding the validation of ORF variants and novel ORFs, we excluded peptides matched to RefSeq ORFs. Therefore, the resulting FLT-specific peptides could not be found for ORF categories including RefSeq annotation, N-terminal truncation and internal deletion. For the remaining categories, we found varied percentages with FLT-specific peptide matches (Fig. 6a, Supplementary Fig. 7a). In particular, in the categories N-terminal extension, internal insertion and novel ORFs in introns, the percentage of ORFs with supporting peptides were comparable to that of the annotated ORFs. Comparing the ORFs with supported peptides to those without in each category, the peptide-supported ORFs had significantly higher ORFscores (Fig. 6b). This is expected because mRNAs with higher ORFscores would likely have higher ribosome occupancy and/or a longer ORF length, resulting in a higher number of detectable peptides. Furthermore, within the peptide-matched ORFs, the number of peptide spectrum matches (PSMs), matched peptides, and unique peptides all positively correlated with the ORFscore (Supplementary Fig. 7b–d), implying that the MS sampling depth might be a bottleneck for supporting ORFs with low expression level and/or translational efficiency. Thus, obtaining deeper MS-based proteomics data specific for low-abundance proteins could further increase the percentage of peptide-supported ORFs.

We then looked further into the novel ORFs detected in introns and RefSeq-annotated UTRs. Among the peptide-supported novel intronic ORFs, we found that their mouse and human homologs were often annotated as separate genes with distinct promoters, again demonstrating the incomplete rat genome annotation. More interestingly, peptide evidence was also found for novel ORFs located in the 5′UTR and 3′UTR regions. If both the RefSeq annotated ORF and these UTR encoded ORFs were translated from the same RNA transcript, they were termed as bicistronic. This phenomenon is frequently observed in bacteria, but is very rare in higher eukaryotes. However, in the gene *Rpp14* (encoding a subunit of RNase P), where we only detected peptides supporting an unannotated ORF in its 3′UTR (Fig. 6c; annotated spectra for the detected peptides shown in Supplementary Fig. 8), an examination of its mouse and human homologs revealed two distinct ORFs encoding proteins with different functions–the protein HTD2, translated from the 3′ UTR of *RPP14*, is a mitochondrial dehydratase. Another example is the gene *Nudt13* (Nudix hydrolase 13), where we detected peptide evidence for both its main ORF and its 3′UTR ORF (Fig. 6d; annotated spectra for the detected peptides shown in Supplementary Fig. 8). In contrast to the highly conserved main ORF, the 3′UTR ORF is not conserved in other species. Whether its function is specific to rat awaits future studies.

**Compartment-localized FLTs in neurons**. As protein translation occurs in both neuronal cell bodies and processes[55], we next investigated which RNA isoforms were transported to neuronal compartments. For this purpose, we divided rat hippocampal slices into somata and neuropil compartments by microdissection, and profiled their transcriptomes using RNA-seq. Isoform expression levels in the two compartments were quantified and compared for neuronal genes[23] in our FLT collection. Among 1,135 multi-isoform neuronal genes, we found 97 isoforms were significantly enriched in the neuropil whereas 82 isoforms were significantly enriched in the somata (Fig. 7a). As an example, we detected an unannotated isoform in the neuropeptide-encoding gene *Tac3* that was enriched in the somatic transcriptome (Fig. 7b). In another example *Bin1*, a gene expressed in many tissues and associated with the Alzheimer's disease[56], we observed that two of its five isoforms were specifically localized to somata or neuropil (Fig. 7c). Interestingly, the skipping of exon 7 was coupled with the complete exclusion of exons 13–16, and this coordinated splicing may play dual roles, including determining the isoform localization pattern and encoding two different protein isoforms with potentially distinct functions. On average, there was no significant difference in their expression levels between neuropil-localized and somata-enriched isoforms (Fig. 7d), indicating that, in general, transcript localization might not be coupled with their expression levels.

## Discussion

In this study, we provided a high-quality resource for gene annotation in the rat hippocampus. Comparing to the RefSeq (1.08 isoforms per gene) and Ensembl (1.43 isoforms per gene) rat annotation, our FLTs' diversity of 3.91 isoforms per gene is more than twofold higher. However, this should be considered as a conservative estimation of transcript isoform diversity. On one hand, some of the PacBio transcripts we discarded could still be in full-length, but their expression levels might be too low to have detectable CAGE and 3′-seq clusters; on the other hand, our approach does not cover extremely long transcripts in full length. This problem is evident when we compared the percentage of genes covered by our FLT transcripts across different lengths (Supplementary Fig. 9): genes longer than 6000 nt were covered with a much lower probability, likely due to the fall-off of reverse transcriptase during the reverse transcription and/or polymerase in the PacBio sequencing. Another factor that impacts the FLT coverage is the transcript abundance. The cDNA normalization applied in this study has largely eliminated this effect and many transcripts of very low abundance were indeed recovered in full length. These transcripts could be potentially transcriptional noise present in all cells, or functional transcripts expressed only in specific cells. Considering the latter possibility, combining our approach with enrichment of distinct cell types or even a single-cell RNA-seq strategy, as reported recently[35], would further expand our understanding of transcriptome diversity.

Based on our FLT data, we investigated the co-occurrence of alternative RNA processing events at a genome-wide scale. The observed co-occurrence of alternative transcript ends is in concordance with the phenomena reported in yeast, where promoters can drive alternative polyadenylation mediated by RNA binding proteins[57,58] or gene loops[59]. Alternatively, transcriptional activity resulting from alternative promoters might also affect the polyadenylation site choice[60]. Among the internal alternative

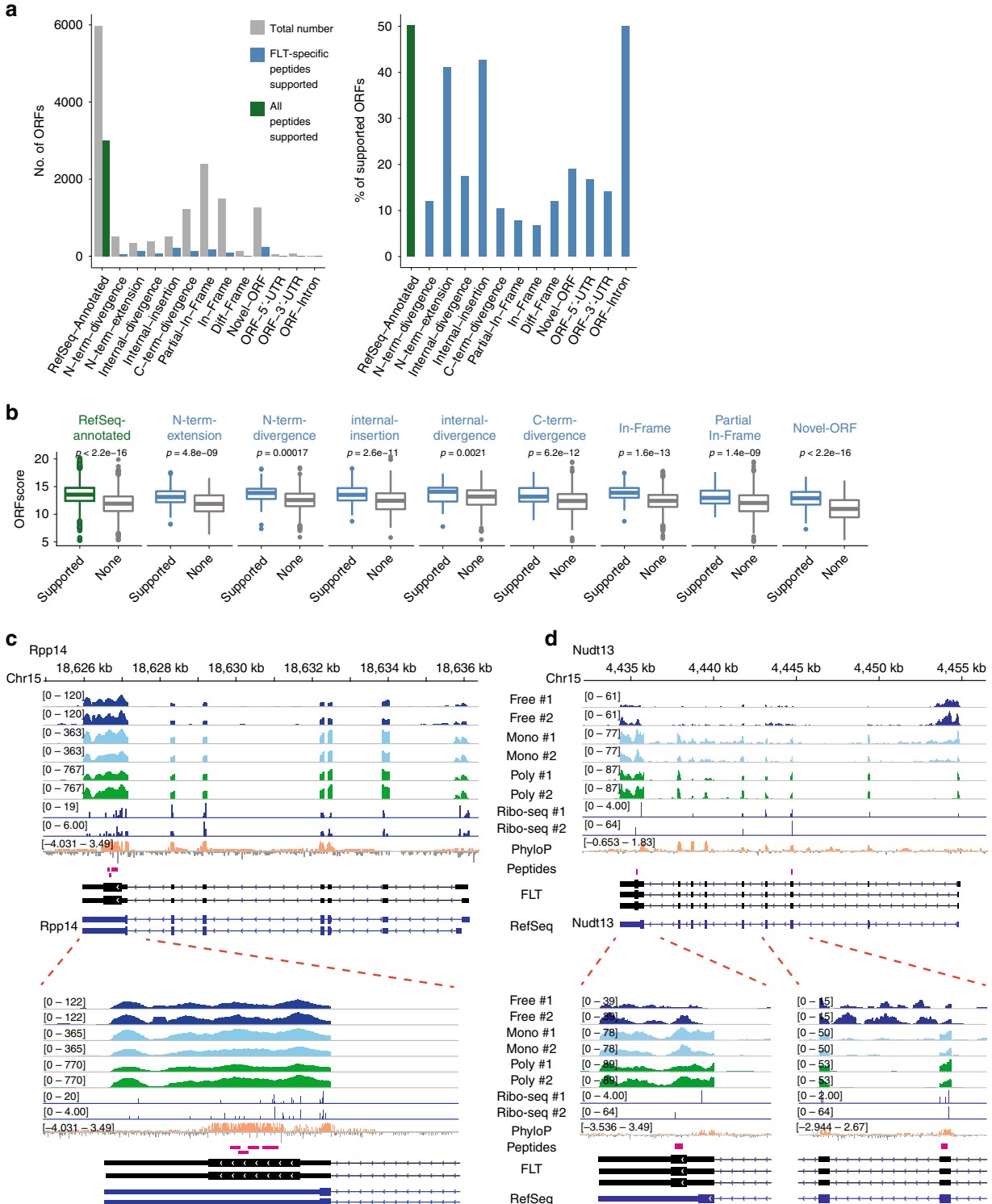

**Fig. 6** Validation of the FLT-based ORFeome using MS-based proteomics. **a** Bar plots show the number (left) and percentage (right) of ORFs supported with peptide evidence. Except for the RefSeq-annotated ORFs where all peptides were used, in the other ORF categories only the FLT-specific peptides were used. N-terminal truncation and internal deletion were excluded since no FLT-specific peptides were expected. **b** Boxplots compare the ORFscores of ORFs with peptide evidence to those without in each category. *P* values (Mann–Whitney *U* test) are indicated above the boxplots. Box edges represent quartiles, whiskers represent extreme data points no more than 1.5 times the interquartile range. **c**, **d** Two examples of peptide-supported novel ORFs identified in the known gene loci. Tracks from top to bottom are genomic coordinates, two replicates of ribosome-free RNA, two replicates of monosome-associated RNA, two replicates of polysome-associated RNA, two replicates of ribosome footprinting, PhyloP conservation score, peptide evidence, full-length transcripts, and RefSeq genes. Zoomed-in regions are also shown at the bottom, with the same track orders. **c** In the gene *Rpp14*, peptide evidence was only found in the novel ORF in its 3′UTR, whereas **d** in the gene *Nudt13*, peptide evidence was found for both the RefSeq-annotated main ORF and the novel ORF in its 3′UTR. Source data for panel **b** are provided in a Source Data file

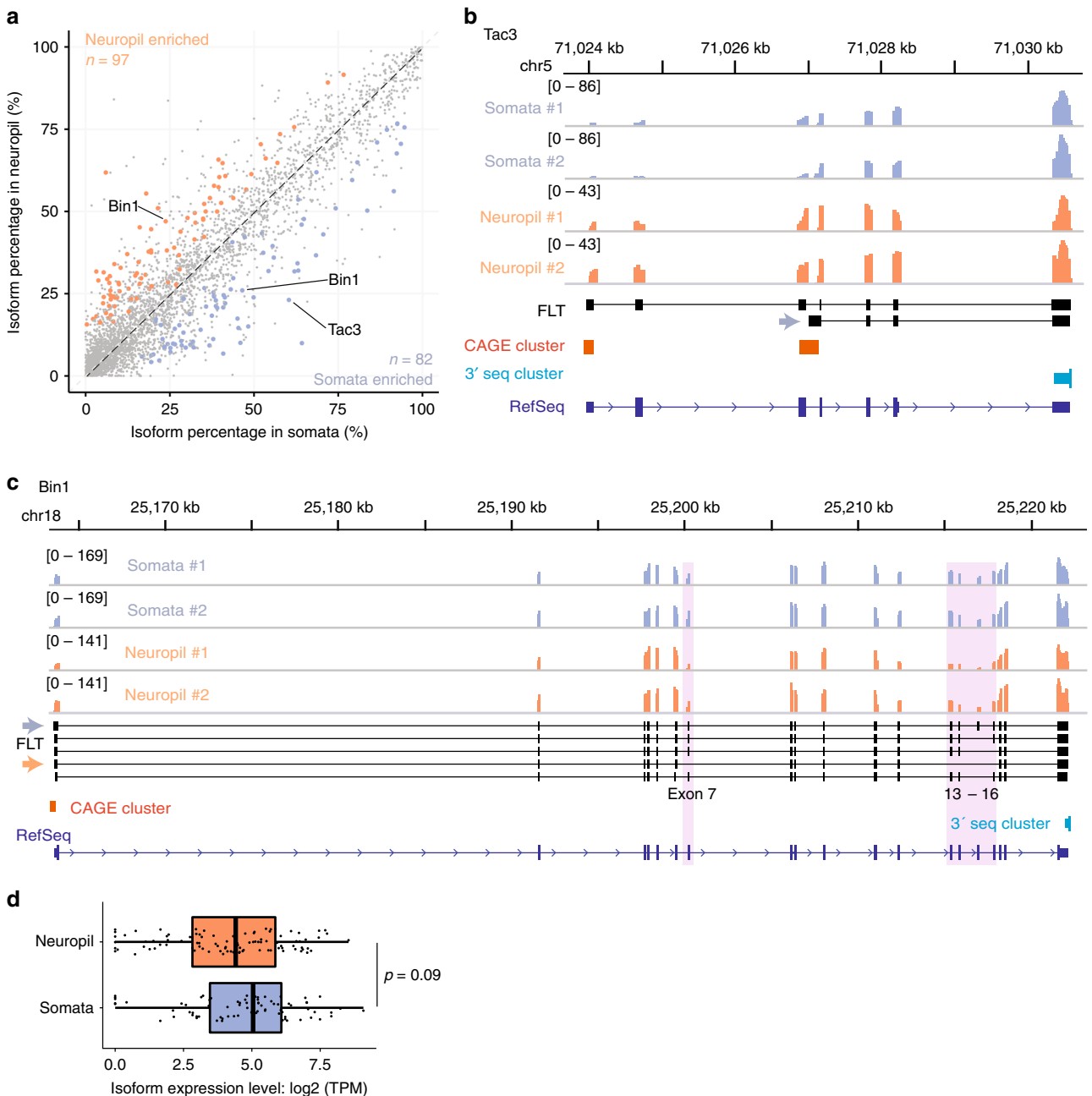

**Fig. 7** Compartment-localized isoforms in neurons. **a** Scatter plot compares the isoform abundance between somata (*x*-axis) and neuropil (*y*-axis) compartments. Significantly localized isoforms were marker with blue (somata) or orange (neuropil) colors and example isoforms were highlighted. **b**, **c** Examples of the compartment-localized isoforms. Tracks from top to bottom are: genomic coordinates, Illumina read coverage in two replicates of somata samples and two replicates of neuropil samples, our FLT isoforms, CAGE clusters, 3′seq clusters, and RefSeq isoforms. Compartment localized isoforms were indicated by arrows. The two genes are: **b** *Tac3*, a neuropeptide-encoding gene, and **c** *Bin1*, a gene expressed in many tissues and associated with the Alzheimer's disease. **d** Boxplots compare the expression levels (log2 TPM) between neuropil and somata localized isoforms (*p* value = 0.09, Mann–Whitney *U* test). Box edges represent quartiles, whiskers represent extreme data points no more than 1.5 times the interquartile range. Source data for panels **a** and **d** are provided in a Source Data file

splicing events, the co-occurrence of alternative donor and acceptor splicing sites indicates that the choice of upstream splice donor sites could influence the selection of downstream splice acceptor sites. Whether this coordination is mediated by the same trans-acting regulators warrants further studies. Finally, the coupling between multiple retained introns indicates that splicing out one intron may apply to other introns in the same gene, suggesting that splicing efficiency is generally regulated in a gene-specific manner[61]. Given that intron retention functionally

downregulates gene expression levels[62], the re-occurrence of intron retention in the same gene could possibly reassure such regulatory effect.

As discussed previously[18], polysome-seq and ribo-seq complement each other. Here, we first used polysome-seq to predict isoform-specific translational status, and then ribo-seq 3-nt periodicity feature to recognize ORFs in isoforms with active translational status. In this way, we largely avoided false positive ORF predictions, which could be caused by shared ribosome

footprinting signals in the overlapped regions between translationally active and inactive isoforms. Indeed, using ribosome footprinting data, ORFs could be detected in 98.7% of the translationally active isoforms defined by the polysome-seq, demonstrating the success of our strategy. When we used random combinations of alternative exons to mimic the ambiguous transcriptome assembly, such transcript sets often yield shorter ORFs and/or ORFs with lower ORFscores (Supplementary Fig. 5c, d). Therefore, the non-full-length transcript annotation can dramatically overstate the proportion of non-coding isoforms and underestimate the ORF diversity.

So far, alternative RNA processing has been extensively studied at RNA level, whereas the analysis at protein level is rather limited. This is mainly due to the technical challenges in measuring multiple protein isoforms: Standard MS-based proteomics involves digesting proteins into peptides, so that the information from which protein isoform a given peptide was derived is lost. As a result, only isoform-specific peptides can be used to unambiguously support ORF variants. Currently, even deep proteomic datasets can just cover a minor fraction of all existing peptides and many isoform-specific peptides escape detection. Therefore, only a small proportion of alternative ORFs have been supported with peptides, and to what extent the RNA isoform diversity can be translated into protein diversity remains unclear[27,28]. In this study, of all the transcript isoforms with sufficient abundance, 31.8% were predicted as translationally inactive. In the remaining translating ones, 21.8% differ only in UTRs whereas 78.2% encode distinct ORFs. As a result, a RNA diversity of 3.91 transcript isoforms per gene dropped to a protein diversity of 1.99 ORFs per gene. We validated a large number of ORF variants and even novel ORFs by MS-based proteomics data. In particular, nearly half of the novel intronic ORFs, the ORFs with internal insertions or extended N-terminus were matched with peptides, comparable to the RefSeq annotated ORFs. As expected, ORFs with peptide evidence were biased towards those with higher ORFscores, which is proportional to the expression level, translational efficiency and the length of the ORF. This again demonstrates the limited sensitivity of MS-based methods, and indicates that deeper proteome sampling would be required to gain further supports for predicted ORFs, particularly those of lower abundance.

Finally, based on our FLTs, we found hundreds of isoforms with distinct localization pattern in neurons. Of note, we very likely understated the number of the compartment-enriched isoforms, because we had to stringently control false positives given the high uncertainty in inferring isoform expression level based on short-read RNA-seq data. In terms of the mechanisms and functions, it still awaits future studies how distinct transcripts are chosen to localize and what roles these transcripts play at distinct subcellular compartments.

## Methods

### Rat hippocampus samples
The procedures involving animal treatment and care were conducted in conformity with the institutional guidelines that are in compliance with national and international laws and policies (DIRECTIVE 2010/63/EU; German animal welfare law; FELASA guidelines). Sprague Dawley rats were housed in standard cages and fed standard lab chow and water ad libitum. Rat hippocampal slices (500 μm) were prepared from four-week-old male animals[23]. The somatic and the neuropil layer of the CA1 region were microdissected carefully by hand from each slice[44]. Tissue pieces were first collected in RNA*later* to stabilize and protect RNA from degradation.

### Total RNA Illumina sequencing
Total RNA from rat hippocampus samples was extracted using TRIzol reagent (Life Technologies) following the manufacturer's protocol. Truseq Stranded mRNA sequencing libraries were prepared with 500 ng total RNA according to the manufacturer's protocol (Illumina). The libraries were sequenced in single-end 1 × 150 nt manner on HiSeq 2000/2500 platform (Illumina).

### PacBio library preparation and sequencing
Starting from 700 ng total RNA, polyA+ RNA selection was performed using Dynabeads® Oligo (dT)$_{25}$ (Invitrogen). The polyA+ RNA was converted to cDNA using the Clontech SMARTer kit for first-strand cDNA synthesis. The resulting cDNA was normalized using a duplex-specific nuclease (DSN)[43]. In brief, the DSN normalization is based on the denaturation-reassociation of double-stranded cDNA coupled with the degradation of the double-stranded cDNA fraction formed by abundant transcripts[63,64]. Then, PCR amplification of cDNA produced by the Clontech method (50× Advantage 2Polymerase mix) utilized a single primer (Supplementary Table 2) since both ends of the cDNA share a common sequence. The PCR products were split into four sections (QIAEX II® Gel Extraction Kit) according to double-stranded DNA size (i.e., <1 kb, 1–2 kb, 2–3 kb, and >3 kb), and a second round of PCR was done to generate sufficient DNA for template preparation. SMRTbell™ template preparation was done according to the manufacturer's guidelines. Size selected libraries were separately sequenced on PacBio RSII SMRT platform according to the manufacturer's instruction (Magbead mode for fragments >3 kb and Diffusion mode for all other fragment sizes; SMRT Cell v3 with 1 × 180 min movie time). The raw sequencing reads from the PacBio RSII SMRT cells were processed through SMRT-Portal analysis suite (PacBio) to subread sequences for further processing.

### Illumina-based error correction of PacBio sequencing reads
We used two computational tools, IPEC and proovread[39], to correct the sequencing errors in PacBio by taking advantage of the more accurate Illumina sequencing. First, we developed the method IPEC, which stands for Illumina-assisted PacBio Error Correction, to augment the sequence accuracy of PacBio reads by using high quality Illumina sequencing data. The method IPEC is an iterative correction approach with seed-and-extension design. In each iteration, for each PacBio read we search for local alignments with all Illumina reads using bowtie2 (with parameters: --local -D 40 -R 3 -N 1 -L 20 -i S,1,0.50 -k 5000 --no-head --ma 2 --mp 6 --rdg 3,3 --rfg 3,3). Mismatches and indels within the local aligned region at the PacBio read are corrected through voting by the multiple mapped Illumina reads. The error-corrected regions serve as a seed for the local alignment in the next iteration. Iteration stops when the accuracy of PacBio reads can no longer be improved significantly. To ensure the best quality of PacBio read error correction, we also processed our raw data by a comparable error correction method, proovread, which is based on the correction-by-short-read-consensus approach[39]. We ran proovread (version 2.13.10) with default parameters. As proovread requires a large memory capacity when dealing with large datasets[40], we split PacBio reads into batches (every 100,000 PacBio reads per batch) when running proovread. After aligning the PacBio reads corrected by the two methods in parallel, we selected the best alignments (see below) and counted the number of correctly aligned bases to estimate the accuracy of corrected PacBio reads.

### Alignment of PacBio reads to the rat reference genome
Since the PacBio long reads cover multiple exons, we used a splicing-aware aligner GMAP[65] (version 2016–09–23) to align the PacBio reads against the rat reference genome (rn6, downloaded from UCSC Genome Browser), with parameters --sampling 1 -B 2 --min-intronlength 20 -n 0, and with four settings of the kmer size (-k) 9, 11, 13, and 15. Only the best alignments, i.e., the largest alignment with the lowest number of mismatches against the reference genome, were kept. Overall, approximately 40% of the final alignments were contributed by IPEC, and the other 60% came from proovread.

### Transcript collapsing and clustering
To extract a non-redundant collection of RNA transcripts from PacBio reads, we developed an in-house approach with an exon-centric clustering method. In brief, after aligning PacBio reads to the rat reference genome (see above), we kept track of the exon combinations for all PacBio reads mapped to the same putative gene locus. We then collapsed these reads into a non-redundant set. Starting from the PacBio reads with the highest number of exons, reads in the final non-redundant set were kept only if any of the following three criteria was met: (i) the combinations of the exon-exon junctions were not identified in the current set; (ii) 5′ end of the first exon was at least 100 nt away from that of current set; and (iii) 3′ end of the last exon was at least 100 nt away from that of the current set.

### CAGE profiling of transcript 5′ends
To generate the CAGE libraries, 5 μg total RNAs collected from rat hippocampus samples were reverse transcribed using random primers (N15-oligo) tailed with 3′ part of Illumina TruSeq Universal Adaptor sequence (P5). Thereafter, 5′complete single-stranded cDNAs were captured based on a protocol from Takahashi et al.[66] with minor modifications. Cap structure and 3′ ends of all RNAs were oxidized by NaIO4 on ice for 45 min, followed by an overnight biotinylation with a long-arm biotin hydrazide at room temperature. Single stranded RNA regions that were not covered by synthesized cDNAs including the 3′ ends were cleaved using RNase I. The 5′ complete cDNAs containing the biotinylated cap site were then captured with Dynabeads® M-280 Streptavidin (Life Technologies). RNAs were hydrolyzed with 50 mM NaOH and single-stranded cDNAs were released from the beads. After ligation with double stranded 5′ linkers with random overhangs (containing 3′ part of Illumina TruSeq Universal Adaptor P7), cDNAs were amplified for 18 cycles using cap forward

primer (containing P5) and cap reverse primer with barcode included. The amplified libraries were sequenced in $2 \times 100$ nt manner on Illumina HiSeq 2000/2500 platform. The primer sequences are listed in Supplementary Table 2.

**CAGE data processing and cluster detection.** The paired-end reads were first subjected to adapter removal using flexbar[67] with the following parameters: -u 2 -m 28 -ae RIGHT -at 2 -ao 1. Then, the first 10 nt of the 1st read was further removed due to potentially high mismatches as derived from the random primer regions. Read pairs that were concordantly mapped to reference sequences of rRNA, tRNA, snRNA, snoRNA, and miscRNAs (available from Ensembl and RepeatMasker annotation) using Bowtie 2[68] (version 2.1.0; in end-to-end and sensitive mode with default parameters) were excluded. The remaining reads were then mapped to the rat reference genome (rn6, downloaded from UCSC Genome Browser) using Tophat2[69] (version 2.0.10) with the following parameters --mate-inner-dist 50 --mate-std-dev 20 -N 3 --read-gap-length 2 --read-edit-dist 3 -min-anchor 6 --library-type fr-firststrand --segment-mismatches 2 --segment-length 25 with the input of RefSeq and Ensembl rat gene annotations (downloaded from UCSC Genome Browser and Ensembl FTP, respectively). Reads that were mapped to multiple genomic loci or whose two ends were mapped to different chromosomes were discarded in the following analysis.

CAGE clusters corresponding to transcription start sites (TSS) were detected[19]. In brief, only the 5′ end positions of the 2nd reads of uniquely, concordantly mapped read pairs (termed as tags hereafter) were used for determining TSS cluster. In general, genomic positions with tags beyond local background and within a distance of 20 nt were assigned into one cluster. Here, the local background (bg) for each position was determined by the maximum of (i) local expectation, that is the average tag coverage in the window of 500 nt centered at the position, and (ii) expression background, that is the sequencing depth-normalized RNA-seq read coverage within the window from 500 nt upstream to 1500 nt downstream of the position.

**Polysome profiling.** Prior to lysis, rat brains were collected and snap frozen in liquid nitrogen and stored at –80 °C. Frozen rat whole brains were pulverized under liquid nitrogen and the powder lysed in lysis buffer (1 ml per 50 mg of tissue; 10 mM HEPES pH 7.4, 150 mM KCl, 10 mM MgCl$_2$, 1% NP-40, 0.5 mM DTT, 100 µg ml$^{-1}$ cycloheximide). After lysing, the cells by passing eight times through 26-gauge needle, the nuclei and the membrane debris were removed by centrifugation ($15,682 \times g$, 10 min, 4 °C). The supernatant was then layered onto a 10-ml linear sucrose gradient (10–50% w v$^{-1}$, supplemented with 10 mM HEPES pH 7.4, 150 mM KCl, 10 mM MgCl2, 0.5 mM DTT, 100 µg ml$^{-1}$ cycloheximide) and centrifuged ($160,000 \times g$, 120 min, 4 °C) in an SW41Ti rotor (Beckman). Ribosome-free, monosome, and polysome fractions were collected and digested with 200 µg proteinase K in 1% SDS and for 30 min at 42 °C. RNA from polysome fractions were recovered by extraction with an equal volume of acid phenol–chloroform (pH 4.5), followed by ethanol precipitation. TruSeq Stranded Total RNA libraries were prepared with 500 ng RNA according to the manufacturer's protocol (Illumina). The libraries were sequenced in $2 \times 150$ nt manner on HiSeq 4000 platform (Illumina).

**Polysome profiling data processing.** Similar to total RNA sequencing, the sequencing reads were first subjected to adapter removal using flexbar[67] with the following parameters: -u 10 -m 36 -ae RIGHT -at 2 -ao 1. Then, polysome profiling data were used to estimate isoform expression levels (see Isoform expression level estimation).

**Ribosome footprinting.** We deeply re-sequenced previously generated ribosome footprinting libraries on rat brains[47]. To prepare the libraries, rat brains were lysed in the same way as for polysome profiling (see above). After lysis, ribosome-protected fragments were collected. In brief, cell lysate was treated with RNase I at room temperature for 45 min. The nuclease digestion was stopped by adding SUPERase In™ RNase inhibitor (Invitrogen) and then loaded onto a linear sucrose gradient (10–50%). After ultra-centrifugation, monoribosome was recovered and RNA was isolated as described for polysome profiling (see above). rRNA was removed using Ribo-Zero™ Magnetic Kit (Human/Mouse/Rat) (Epicentre). The 28–32-nt ribosome-protected fragments were purified through 15% wt vol$^{-1}$ polyacrylamide TBE-urea gel. The size-selected RNA was end-repaired by T4 PNK for 1 h at 37 °C. The sequencing libraries were then generated using TruSeq Small RNA Sample Preparation kit (Illumina) and sequenced in $1 \times 50$ nt manner on Illumina HiSeq 2000 platform.

**Ribosome footprinting data processing.** Similar to total RNA sequencing, the sequencing reads were first subjected to adapter removal using flexbar[67] with the following parameters: -u 2 -m 18 -ae RIGHT -at 2 -ao 4. Reads that were mapped to the reference sequences of rRNA, tRNA, snRNA, snoRNA, and miscRNAs (available from Ensembl and RepeatMasker annotation) using Bowtie 2[68] (version 2.1.0; in end-to-end and sensitive mode with default parameters) were excluded. The remaining reads were then mapped to the rat reference genome (rn6, downloaded from UCSC Genome Browser) using Tophat2[69] (version 2.0.10) with following parameters -N 2 --read-gap-length 2 --read-edit-dist 3 --min-anchor 6 --

library-type fr-secondstrand --segment-mismatches 2 --segment-length 20 with the input of RefSeq and Ensembl rat gene models. Reads that were mapped to multiple genomic loci were discarded in the downstream analysis.

**Isoform expression level estimation.** After processing sequencing reads with flexbar[67], we applied RSEM[70] to estimate the expression levels of each isoform in a gene. RSEM analysis included two major steps: (i) reference transcriptome sequence index building, and (ii) read mapping and expression level inference. In the first step, the sequences in our FLT was fed to the command line tool rsem-prepare-reference. In the second step, with the command line tool rsem-calculate-expression and arguments --paired-end --bowtie2, we went through bowtie2 mapping, alignment parsing, and isoform expression estimation by EM algorithm.

**Translationally active isoform prediction.** Mainly based on the polysome profiling data, we trained a set of support vector machine (SVM) classifiers to predict the translational status of each isoform in our FLT. SVM is a widely-used supervised machine learning approach, which tries to find a soft-margin hyperplane in high-dimensional space shaped by support vectors to classify binary samples. To build the SVM classifiers we took all expressed Ensembl isoforms as the training set, and their translational status was annotated as transcript_biotype. Compared to RefSeq, the Ensembl annotation contained a slightly higher number of isoforms (Fig. 3a) and collected more non-coding transcripts (though many of them were only based on computational predictions). We sorted Ensembl transcripts into two classes regarding their translational status. The biotype protein_coding was taken as translationally active, and others including lincRNA, retained_intron, processed_transcript, nonsense_mediated_decay, and non_stop_decay as inactive. Although the translationally inactive class included five biotypes, there were ~10-fold more isoforms in the active class. To balance the sample size in the two classes, we randomly subsampled the active isoforms. We repeated the subsampling 21 times, resulting in 21 pairs of inactive and active isoform sets. Then we built one SVM classifier in each pair of two classes of samples: We extracted 13 features for each sample (see below), chose the radial basis function as the kernel function of SVM, and the parameters were optimized by grid search using the grid.py script in 10-fold cross validation (CV) mode. The above procedure gave us 21 classification models, and the averaged 10-CV classification accuracy was 86.31%.

Out of the 28,268 isoforms in our FLT, 3712 were excluded for classification, because the estimated expression level of these isoforms in all the three polysome-profiling fractions was close to zero. For the remaining isoforms, the same features as used in SVM training were extracted, which included 12 properties extracted from the polysome profiling data, i.e., isoform expression levels in the three fractions in two replicates ($2 \times 3$), and ratios of isoforms abundance levels between every two out of the three fractions in two replicates ($2 \times 3$). In addition, the length of the longest possible ORF in each isoform was chosen as the 13th feature. Then the 21 trained SVM classifiers were applied to predict the translational status of the isoforms in our FLT, and the final prediction labels were given by classifier voting.

**Translating ORF detection.** To determine the translating ORFs for the translationally active isoforms, we mainly used ORFscore, which was calculated from the ribosome footprinting data. By post-processing ribosome footprinting reads, we found only reads of length 28 nt, 29 nt, and 30 nt showed the typical 3-nt periodicity in annotated ORFs, and the +12 nt position from the read 5′end was corresponding to the P-site of the ribosome[48]. After designating each read as the +12 nt position, we counted the number of reads mapped to each of the three reading frames of one possible ORF, and calculated the ORFscore as follows[48]:

$$\text{ORFscore} = \log_2\left(\left(\sum_{i=1}^{3} \frac{(F_i - F')^2}{F'}\right) + 1\right) \times \begin{cases} -1 & \text{if} (F_1 < F_2) \cup (F_1 < F_3) \\ 1 & \text{otherwise} \end{cases} \quad (1)$$

where $F_i$ is the number of reads in frame $i$, and $F'$ is the average number of reads across all the three frames. We defined translating ORFs as ORFs with ORFscore > 5; in cases where multiple ORFs with ORFscore > 5 in one isoform, these ORFs were prioritized by their length.

**Comparative analysis of isoform and ORF diversity.** The number of gene loci in our FLT comprising different numbers of transcript isoforms and ORFs was counted, respectively. To compare the diversity to that of RefSeq and Ensembl annotation, only the set of gene loci that overlapped with our FLT collection were considered. To facilitate the analysis of alternative RNA processing events, we developed a software toolkit named FuLeTA, available at https://github.com/sunlightwang/FuLeTA.

**Peptide search for ORF validation.** To evaluate the reconstruction of the ORFoeme based our FLT, bottom-up proteomics data from primary hippocampal neurons[54] were re-analyzed. The raw data are available via PRIDE with accession number PXD008596 (untreated/t0 data files). Using the Thermo Proteome Discoverer™ (PD) software suite (version 2.3) with the MS-Amanda identification algorithm[71], these data were matched to a protein database based on 18,015 RefSeq annotated ORFs, a database derived from our FLT-based ORFoeme comprising 10,775 sequences after excluding identical ones in RefSeq, and additionally a

database containing common MS-contaminations. The searches allowed a precursor mass tolerance of 5 ppm and a fragment mass tolerance of 0.02 Da. Carbamidomethylation of Cysteine was set as a fixed modification and oxidation of Methionine was set as variable modification. The parameters further included tryptic peptides with up to two missed cleavages and a minimum of one unique peptide per protein group. Percolator node, a machine learning tool in PD 2.3, was used to estimate the number of false positive identifications[72]. A high confidence $q$-value (i.e., false discovery rate) threshold of 0.01 was assigned to filter both the peptide spectrum match (PSM) results and peptide results. We estimated the FDR on the combined RefSeq/FLT database and also separately for the RefSeq and FLT databases. At combined FDR 1%, we matched 56,233 and 2557 peptides in the RefSeq and FLT databases, respectively; at split FDR 1%, we identified 53,684 and 2175 peptides in the RefSeq and FLT databases, respectively.

**Reporting summary**. Further information on research design is available in the Nature Research Reporting Summary linked to this article.

## Data availability

All the raw sequencing data along with processed data are deposited into NCBI GEO database with accession number GSE128136. MS-based proteomics data are available via PRIDE with accession number PXD008596 (untreated/t0 data files). The source data underlying Figs. 1b, 2b, 2c, 3a, 3d, 4c, 5b, 6b, 7a, and 7d and Supplementary Figs. 1c–e are provided as a Source Data file. All other data are available from the corresponding authors on reasonable request.

## Code availability

Illumina-assisted PacBio Error Correction (IPEC) software is available at https://github.com/arthuryxt/IPEC, and the toolkit for full-length transcriptome characterization is provided at https://github.com/sunlightwang/FuLeTA for free for academic use. Other analysis scripts/codes are available upon request.

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

## Acknowledgements
This work was supported by an Exploration Grant of the Boehringer Ingelheim Foundation (BIS), Sino-German (NSFC-DFG) Cooperative Research Project (No. 31861133013), National Natural Science Foundation of China (No. 31771443), the Basic Research Grant from Science and Technology Innovation Commission of Shenzhen Municipal Government (No. JCYJ20170307105752508), Shenzhen-Hong Kong Institute of Brain Science-Shenzhen Fundamental Research Institutions (2019SHIBS0002). E.M.S is funded by the Max Planck Society and by European Research Council (ERC) under the European Union's Horizon 2020 research and innovation program (grant agreement No 743216). This work was also partially supported by Center for Computational Science and Engineering of Southern University of Science and Technology for providing the high-performance computational infrastructure.

## Author contributions
W.C., X.W. and X.Y. conceived the project. X.W. and X.Y. analyzed all the sequencing data. J.D.L., F.R. and G.T. performed proteomics data analysis. J.H. performed polysome profiling and CAGE profiling, and C.Q. performed PacBio library preparation and sequencing. I.V. and I.E. provided the rat samples. G.T., B.S., W.S., L.F., G.L. and Y.H. helped with the experiments and data analysis. X.W., W.C. and E.M.S. wrote the paper with input from X.Y., J.H., J.D.L. and C.Q. All authors contributed to the edits and comments on the paper.

## Competing interests
The authors declare no competing interests.

## Additional information

**Peer Review Information** *Nature Communications* thanks Nuno Barbosa-Morais and other, anonymous, reviewer(s) for their contribution to the peer review of this work. Peer reviewer reports are available.

