## [Peer Review File · Nature Communications]

Reviewers' comments:

Reviewer #1 (Remarks to the Author):

This manuscript presents full-length transcript annotations derived from rat hippocampal RNA, along with an analysis of the translational status of these isoforms. Transcripts are reconstructed from a combination of short-read and long-read sequencing; independent 5'- and 3'-end capture data are used to validate full-length transcripts and in particular to exclude 5' truncations. This approach provides four isoforms per gene, on average. These reflect a variety of alternative initiation, splicing, and 3' end formation events consistent with patterns seen in other mammalian transcriptomes. The translational status of these transcripts is assessed through further sequencing of polysome-associated transcripts combined with the analysis of existing ribosome profiling data. Transcriptome diversity is translated into proteome diversity, and many genes produce more than one protein isoform. Novel reading frames are reported as well. These data are further intersected with proteomics data that provide some support for alternative translation products, especially extensions and insertions that are expected to yield a substantial repertoire of novel peptides. Finally, it is reported that some isoforms show differential localization in RNA-Seq samples from anatomically distinct regions of the hippocampus.

The annotations and analyses reported here provide a valuable resource for the community, and the manuscript integrates a wide variety of data. Specific and potentially interesting examples of transcript diversity are highlighted as well.

I do have a few concerns, however:

1. The analysis relies in part on a tool for combining Illumina and Pac Bio reads, called IPEC, that appears to be unpublished. There isn't any validation of the accuracy of IPEC or the quality of the corrections.
2. As a related point, the alternate Illumina + Pac Bio correction tool, "proovread", is validated to a limited extent in the comparative review [ref 63] because the review authors could not run it on large data sets.
3. To what extent do Illumina-based corrections of Pac Bio transcripts "fix" unalignable reads or structural errors? That is, how different are the results of Pac Bio to genome alignment after IPEC/proovread correction?

4. Are highly similar genes or pseudogenes a concern when “correcting” Pac Bio reads with Illumina sequencing? That is, could Illumina reads from a distinct but very similar genomic locus map to a Pac Bio read?

5. The phrase “inactive translation” used to refer to transcripts that don’t associate with polysomes is a bit confusing — it sounds as if it’s referring to paused ribosomes or something like that, because “inactive” modifies the “translation”. Consider “translationally inactive” or “non-translated”.

6. I found the analysis of artificially constructed transcripts with randomized alternate exons (ll. 332 - 338) unclear. Given that nonsense-mediated decay should rapidly eliminate out-of-frame alternate transcripts, should the randomization consider only frame-preserving variants?

7. In the discussion, it is noted that intron retention appears coupled (l. 508). In fact, the presence of some amount of nuclear pre-mRNA will give the appearance of coupled intron retention.

8. The axes in Supplementary Figure 1a compress the relevant size ranges. Could all “long” transcripts be collapsed into a “>5kb” category or similar, allowing better resolution of the 1 - 5 kb range that is spanned by the size selection?

9. In the context of the present paper, it seems important to discuss the observation that frame-preserving microexons that alter protein-coding sequences are highly enriched in neurons (Irimia et al., Cell 2014).

Reviewer #2 (Remarks to the Author):

In their manuscript Wang et al reconstruct full length transcriptome of rat hippocampus. The authors used hybrid approach combining short-read and long-read transcriptome sequencing techniques. They complimented transcriptome data with CAGE and 3P-Seq data to fine-map precise start and end points of transcripts. They predicted open reading frames (ORFs) using polysome-seq and ribosome footprinting data and confirmed some of the new ORFs with previously obtained mass-spec data. Finally the authors investigated sub-cellular localization of isoforms in neurons stomata and neuropils.

This is a very well designed study and well-written manuscript. The authors use comprehensive approach (cDNA normalization, size fractionation, long-read error correction using Illumina reads). Using their approach they annotated 28,268 transcripts for 6,380 RefSeq genes as well as 849 previously un-annotated loci. They investigated co-occurrence of alternative transcript features and, based on this analysis, report quite some interesting observations.

The presented set of full-length transcripts will be useful resource for improving annotation of rat genome, as a test set in evaluation of full length transcripts reconstruction and isoform quantification using short-read sequencing.

I read the manuscript with great interest and did not find any major drawbacks. I recommend it for publication in its current form.

Minor point:

Line 81: coding sequencing -> coding sequences

Reviewer #3 (Remarks to the Author):

This manuscript describes a hybrid workflow for reconstructing the rat hippocampal transcriptome, by taking advantage of the ability to sequence full-length transcripts offered by PacBio, while rectifying its relatively high error rate by correcting the sequences of its long reads through local alignments with short low-error Illumina sequencing reads. Moreover, comparison with independent 5' and 3'-end profiling allowed for the definition of reliable full-length transcripts, while integration with polysome/ribosome-profiling data enabled the prediction of actively translating isoforms. This hybrid workflow resulted in an increase in the number and accuracy of transcripts annotated in the rat genome, providing an important contribution to the characterization of the rat hippocampus transcriptomic and proteomic diversity. The manuscript's Introduction is also a very useful concise and up-to-date summary on the recent advances in sequencing technologies for studying the transcriptome, including the limitations of "third generation sequencing".

Overall, the article is very well written, the results of the work described therein are of great interest to the scientific community and we have no major concerns about its soundness.

We do, however, request that the following issues are addressed by the authors for the sake of improving clarity and preventing mis- and over-interpretations:

1. [Introduction, page 4, line 88] The concept of “coordinated alternative splicing” is not introduced and may not be familiar to the broad readership of Nature Communications. given its biological relevance and the usefulness of the described methodologies in its study, coordinated alternative splicing should be explicitly (even if very briefly) defined.
2. [Results, page 7, 1st paragraph, lines 140-141, and Figure 1a] Panel 1a does not visually discriminate the stated “experimental and computational arms”. The common reader will need to resort to the figure’s caption to learn this distinction. Accordingly coloring the flow chart’s rectangles would greatly facilitate the intended distinction.
3. [Results, page 7, 2nd paragraph, lines 147-154] The authors unrealistically assume specialized knowledge from readers. Why were libraries size-fractionated and how are cDNA sizes expected to impact the accuracy of PacBio sequencing? Moreover, the concept of “subread” is not introduced – this can perhaps be done in the Introduction when talking about CCSs. Finally, “in general, the subread length in the four libraries” showing “a tendency to match the expected size” is an overstatement. For larger sizes (3rd and 4th panels of Supplementary Fig. 1a) it is more than “a fraction of reads” “outside of expected size ranges”.
4. [Results, page 7, 3rd paragraph, lines 156-164] The comparability between PacBio reads-per-gene and Illumina’s RPKMs is not entirely clear. Would plotting one against the other for the same genes result in something similar to Figure 1b? Given that RPKMs involve correction for transcript length (fewer reads for shorter transcripts, given the same expression level), does this variable introduce a bias contributing to normalization discrepancies?
5. [Results, page 9, lines 189-191, and Figure 2a] For clarity in the interpretation of the figure, it should be made explicit that the absence of a CAGE “ground truth” 5’ end explains the invalidated transcript.
6. [Results, page 11, lines 261-263, and Figure 3d and Supplementary Fig. 3c] The observed significant depletion of co-occurrence of intron retention and exon skipping should be revisited distinguishing alternative exons that fall within or neighbor introns with evidence for retention and those that do not, to control for the necessary mutual exclusivity of the two types of events when the associated exon and intron are adjacent. Also, given the reported increase in intron retention in non-neuronal genes during neuronal development (e.g. PMID 25258385), it would be very insightful to discriminate between introns in neuronal and non-neuronal genes and check if the co-occurrence of multiple intron retention events is different between these two groups.

7. [Results, page 13, lines 297-300] This sentence suggests that exon skipping, intron retention, alternative 3' and 5' splice sites and alternative last exons are generally ORF-disrupting, which is not the case (e.g. most alternative exons keep the frame, with their length being a multiple of 3nt). Rephrasing is recommended in order to avoid imprecise interpretations of the conveyed message.
8. [Results, page 14, lines 332-338] Rephrasing of this paragraph is also recommended. For instance, it is not clear what "appropriately coordinated RNA processing" means in that context. Deeming as "real" the isoforms observed in the active translation pool is somewhat misleading, as it suggests a ground truth and that others are "not real". Similarly, talking about "properly programmed RNA processing" suggests that alternatively processed isoforms are somewhat "not proper"... For instance, how does programmed RNA processing associate with ORF length? Is the "properly" programmed RNA processing isoform necessarily that producing the longer ORF?
9. [Discussion, page 20, 1st paragraph, lines 460-463] It is not clear why the dependence of tools on gene annotation is "increasing".
10. [Discussion, page 21, lines 484-487] Where are the data for the comparison of "the percentage of genes covered by our FLT transcripts across different lengths"?
11. [Discussion, page 21, lines 506-508] Stating that "co-occurrence of alternative donor and acceptor splicing sites indicates that the choice of upstream splice donor sites could influence the selection of downstream splice acceptor sites" is almost a truism and not very insightful. Mentioning at least the hypothesis that alternative donor and acceptor sites could be regulated by the same trans-acting splicing factors would provide more food for thought for the splicing community.
12. [Discussion, page 22, lines 508-510] Stating "that the defect in splicing out one intron may apply again to other introns in the same gene" implies that splicing efficiency tends to be gene-specific rather than intron-specific. This is a bold hypothesis that needs to be elaborated on. Moreover, taking intron retention as a splicing "defect" overlooks its regulatory role (e.g. PMID 25258385). Rephrasing and/or referencing is therefore advised.
13. Figure 1d would greatly benefit for having the red/grey color code in a legend inside the panel and not only in the caption.

We also suggest the following minor "proofreading" edits:

14. Figure 1c: there is a mismatch between the colors in the legend and the colors in the histograms themselves that make their interpretation quite tricky. The authors should look for a clearer and consistent way of discriminating the two distributions.
15. The percentages of RefSeq-overlapping isoforms in Figure 3B add up to 101% and those of multiple alternative events to 62%, and not the 61% mentioned in line 249 (page 11). This is clearly just a "rounding" issue that should be fixed to prevent confusion.
16. "Subread" is used in line 147 and "sub-read(s)" is used in line 617 and Figure 1c.

17. PacBio “reads-per-gene” is used in line 158 and “reads per gene” in line 1202.
18. Line 301, page 13: “significantLY” instead of “significant”.

Reviewer #4 (Remarks to the Author):

1. In the MS data analysis, the authors used a database derived from the FLT-based ORFoeme and a database based on RefSeq annotated ORFs for database searching. It looks like a combined database was used for the search, and the FDR estimation was performed without distinguishing novel and annotated peptides. It is well established that such analysis will underestimate the true FDR for novel peptides. The authors will need to perform separate FDR estimation to ensure the quality of the identified novel peptides.
2. To further illustrate the quality of the novel peptide identifications, the authors should compare the score distribution for novel peptides against that for annotated peptides. Either MS Amanda score or Percolator score can be used for the analysis.
3. Was SILAC related modifications considered in database searching?
4. Please report the numbers of protein sequences in the RefSeq and novel databases in the method section.
5. How many annotated and novel peptides were identified under 1% peptide level FDR?
6. Please provide a table for all the identified novel peptides as a supplementary table, including spectrum title, m/z, charge, score, q-value, precursor delta mass, peptide sequence, modification and protein ID.
7. Please provide annotated spectra for the peptides presented in Figure 6.
8. Why the analysis only used “untreated/t0” data files from dataset PXD008596 instead of data from all time points?
9. I am confused about the active translating isoform prediction section. First, in the main text, it mentioned that “out of the 28,268 isoforms in our FLT dataset, 3,712 were excluded” but in the method section, “3,819 were excluded”. Second, it looks like the 28,268 isoforms were used for model training based on Ensembl annotation, and then the model was used to classify the same set of isoforms. How many isoforms were annotated by Ensembl? When there is a disagreement between Ensembl and the model assignment, which one should be trusted?
10. When the authors compared the isoform and ORF diversity of the FLTs to that of RefSeq and Ensembl genes, was the analysis based on the overlapping set of genes?

11. In the analysis of co-occurrence of alternative events, RefSeq was used as the reference, which might be misleading. It is unclear whether each transcript sequence in RefSeq comes from an individual sample in a specific condition.

Point-by-point response to reviewers' comments

Reviewer #1 (Remarks to the Author):

This manuscript presents full-length transcript annotations derived from rat hippocampal RNA, along with an analysis of the translational status of these isoforms. Transcripts are reconstructed from a combination of short-read and long-read sequencing; independent 5'- and 3'-end capture data are used to validate full-length transcripts and in particular to exclude 5' truncations. This approach provides four isoforms per gene, on average. These reflect a variety of alternative initiation, splicing, and 3' end formation events consistent with patterns seen in other mammalian transcriptomes. The translational status of these transcripts is assessed through further sequencing of polysome-associated transcripts combined with the analysis of existing ribosome profiling data. Transcriptome diversity is translated into proteome diversity, and many genes produce more than one protein isoform. Novel reading frames are reported as well. These data are further intersected with proteomics data that provide some support for alternative translation products, especially extensions and insertions that are expected to yield a substantial repertoire of novel peptides. Finally, it is reported that some isoforms show differential localization in RNA-Seq samples from anatomically distinct regions of the hippocampus.

The annotations and analyses reported here provide a valuable resource for the community, and the manuscript integrates a wide variety of data. Specific and potentially interesting examples of transcript diversity are highlighted as well.

Thank you for the positive comments and evaluation of our study.

I do have a few concerns, however:

1. The analysis relies in part on a tool for combining Illumina and Pac Bio reads, called IPEC, that appears to be unpublished. There isn't any validation of the accuracy of IPEC or the quality of the corrections.

The concept of using Illumina sequencing data to augment the accuracy of PacBio sequencing has been widely applied and accepted in previous full-length transcriptome studies and other applications, e.g., LSC¹ and LoRDEC². When we started the analysis of our own datasets six years ago, there were no available benchmarking studies that comparatively evaluate tools for Illumina data assisted PacBio read error correction. Therefore, we tested a set of existing methods (including LSC, LoRDEC, and proofread) on a subset of our data, and found that the software proofread achieved the best performance in terms of overall alignment rate and accuracy at that time. This result generally agrees with a recent review (Fu et al, 2019)³ that proofread is among the best tools in the alignment rate and the accuracy of corrected reads, though its memory demands are a practical issue (see the response to comment #2).

Nevertheless, we still found that in some cases proofread failed to correct sequencing errors in the PacBio data, and thus we started to develop the method IPEC, which is an iterative

algorithm based on a seed-and-extension design. In brief, in each iteration PacBio reads are searched for local alignments of Illumina reads, and the local aligned regions at PacBio reads are corrected through voting by the mapped Illumina reads. Then, the error-corrected regions serve as a seed for the local alignment in the next iteration until the improvement is saturated. By aligning the corrected reads to the rat reference genome, we found that the sequence accuracy was dramatically improved. Comparing IPEC and proofread, there was no clear winner, so we decided to combine their results for a better full-length transcriptome annotation. Taking the best alignment for each read after IPEC and proofread error correction (i.e. the longest alignment with the lowest number of mismatches against the reference genome), we found that whereas approximately 40% of the final alignments were contributed by IPEC, the other 60% were achieved by proofread. In the revised manuscript, we have added/clarified this information in the Methods section (p 27).

In the revision, we have also made the software IPEC publicly available online at GitHub: <https://github.com/arthurxt/IPEC>. Example data and command-line scripts are also provided for users.

2. As a related point, the alternate Illumina + Pac Bio correction tool, “proofread”, is validated to a limited extent in the comparative review [ref 63] because the review authors could not run it on large data sets.

We chose proofread as an alternative tool for PacBio error correction as a result of in-house software comparison (please see above). Regarding the software crashes on large data sets, we indeed found that proofread requires a lot of memory. To resolve this memory issue, we split PacBio reads into batches (every 100,000 PacBio reads per batch), but used all the Illumina reads to augment the sequence accuracy in each batch. As each PacBio read was treated independently, we did not sacrifice the performance of proofread. We have added this information to the revised manuscript (p 26).

3. To what extent do Illumina-based corrections of Pac Bio transcripts “fix” unalignable reads or structural errors? That is, how different are the results of Pac Bio to genome alignment after IPEC/proofread correction?

We agree that the PacBio alignment rate and structural accuracy are critical in our full-length transcriptome annotation. In our original manuscript, we only showed the increased accuracy of PacBio sequencing reads after the error correction. We reasoned that higher sequence accuracy would enhance the alignability and the alignment precision, for example, surrounding splice sites. However, these data were not shown. In the revised manuscript, we have added these data to support this point. In brief, the overall PacBio read mappability increased from 72.8% to 98.2% after error correction (Fig. 1d), the alignment coverage was elevated to different extents depending on associated gene expression levels (Supplementary Fig. 1g), and alignment precision at canonical splice sites was also largely enhanced (Supplementary Fig. 1h).

4. Are highly similar genes or pseudogenes a concern when “correcting” Pac Bio reads with Illumina sequencing? That is, could Illumina reads from a distinct but very similar genomic locus map to a Pac Bio read?

It is challenging to correct PacBio sequencing errors using Illumina data for those derived from genes of high similarity or pseudogenes. This point has not been extensively discussed in the literature. In fact, both our algorithm IPEC and the published tool proofread cannot avoid mapping Illumina reads from a distinct but very similar locus to PacBio reads. Nevertheless, since pseudogenes are usually expressed at much lower levels than their ‘parental’ genes, the voting procedure in error corrections would assure the correction will not be largely biased towards the pseudogenes.

To further investigate to what extent possible pseudogene-derived Illumina reads could affect our error correction results, we took the annotated pseudogenes from Ensembl (1656 gene loci). Out of 41.9 million mappable Illumina reads, only 1.57 million (3.7%) aligned better to Ensembl pseudogenes. The 1.57 million pseudogene-derived Illumina reads were mapped to 14.5% of our PacBio reads using the same parameters as we used for error correction. Then, we compared the sequence accuracy of the 14.5% PacBio reads after error correction to the rest. As shown in Fig R1A, the PacBio reads with pseudogene-derived Illumina reads mapped had a slightly lower accuracy, suggesting that the existence/expression of pseudogenes may have a small impact on the error correction procedure. This impact was apparent only when gene expression was very low (Fig R1B).

Figure R1. Comparison of sequence accuracy after error correction between PacBio reads with pseudogene-derived Illumina reads mapped and the others.

5. The phrase “inactive translation” used to refer to transcripts that don’t associate with polysomes is a bit confusing — it sounds as if it’s referring to paused ribosomes or something like that, because “inactive” modifies the “translation”. Consider “translationally inactive” or “non-translated”.

Thanks for pointing out this potentially misleading phrase. This class of transcripts were predicted using supervised SVM (support vector machine) classifiers. When training the classifiers, we took all non-coding transcripts annotated in Ensembl as known cases of this category. In this sense, we should call them ‘non-translated’. However, according to the most recent research, we cannot be 100% sure that “non-coding” or “non-translated” RNAs do not provide the template for some protein products, in particular small peptides. Therefore, we think that the reviewer’s suggested phrase ‘translationally inactive’ is more appropriate here. In the revised manuscript, we have rephrased the terms accordingly.

6. I found the analysis of artificially constructed transcripts with randomized alternate exons (ll. 332 - 338) unclear. Given that nonsense-mediated decay should rapidly eliminate out-of-frame alternate transcripts, should the randomization consider only frame-preserving variants?

We agree with the reviewer that the out-of-frame transcripts are likely to be quickly degraded via the nonsense-mediated decay (NMD) pathway. However, the purpose of this simulation is to examine how properly coordinated alternative RNA processing shapes the whole transcriptome/proteome, and the out-of-frame transcripts are regarded as improperly processed transcripts. Therefore, it is appropriate to take these transcripts into the ‘background’ set, to which we compare our ‘foreground’ set comprising all detected transcripts resulted from properly RNA processing.

7. In the discussion, it is noted that intron retention appears coupled (l. 508). In fact, the presence of some amount of nuclear pre-mRNA will give the appearance of coupled intron retention.

We agree that the presence of nuclear pre-mRNAs should also lead to the observation of co-occurrence of multiple retained introns. However, in our experiments, we chose only poly-adenylated transcripts for PacBio/Illumina transcriptome sequencing and analysis, which to a large extent excludes the potential contamination of pre-mRNAs. Moreover, we checked the size distribution and the relative positions of retained introns in our FLT collection and found that they were not different from those based on RefSeq annotation, suggesting that pre-mRNAs did not contaminate our FLT collection.

8. The axes in Supplementary Figure 1a compress the relevant size ranges. Could all “long” transcripts be collapsed into a “>5kb” category or similar, allowing better resolution of the 1 - 5 kb range that is spanned by the size selection?

Following the reviewer’s suggestion, we have redrawn the histograms in Supplementary Figure 1a, in which we have increased the overall resolution. However, we did not collapse the long transcripts, as they exist abundantly in the ‘>3kb’ libraries, and it would be interesting to check how they are distributed in a comparable scale. With the natural, linear scale as shown in the current histograms, it is also simpler to interpret the overall length distribution.

9. In the context of the present paper, it seems important to discuss the observation that frame-

preserving microexons that alter protein-coding sequences are highly enriched in neurons (Irimia et al., Cell 2014).

This is a valuable point. Now we have extensively explored our data regarding microexons (exons of 3–27 nt-long, following the definition in Irimia et al⁴), in particular, ORF-preserving microexons. Different from the scope of Irimia et al, where the authors analyzed alternatively spliced microexons comparing across >50 diverse cell and tissue types⁴, here we could only focus on the FLT in the rat hippocampus. Nevertheless, qualitative analysis of isoform configuration in our FLT identifies 141 cassette microexons in 131 genes. Of note, the number must be a large underestimate because microexons appearing in all the FLT isoforms in the rat hippocampus were not considered as cassette exons. Interestingly, among the 141 microexons, 98 (70%) are frame-preserving, largely exceeding the ratio of frame-preserving cassette exons over all cassette exons regardless of their length (41%; Supplementary Figure 2e). Next, we further classified the 131 genes into neuron-enriched, glia-enriched, and non-enriched ones using the 3'end-seq data derived from neuron-enriched and glia-enriched primary cultures⁵, and found a tendency for frame-preserving microexons to be slightly more enriched in neuron-enriched genes compared to non-enriched ones (72% vs. 61%). Both observations indicate that cassette microexons detected in the rat hippocampus have a large potential to generate alternative protein isoforms which are related to neuronal functions.

We have added the result regarding frame-preserving microexons to the revised manuscript (pp 10-11).

Reviewer #2 (Remarks to the Author):

In their manuscript Wang et al reconstruct full length transcriptome of rat hippocampus. The authors used hybrid approach combining short-read and long-read transcriptome sequencing techniques. They complimented transcriptome data with CAGE and 3P-Seq data to fine-map precise start and end points of transcripts. They predicted open reading frames (ORFs) using polysome-seq and ribosome footprinting data and confirmed some of the new ORFs with previously obtained mass-spec data. Finally the authors investigated sub-cellular localization of isoforms in neurons stomata and neuropils.

This is a very well designed study and well-written manuscript. The authors use comprehensive approach (cDNA normalization, size fractionation, long-read error correction using Illumina reads). Using their approach they annotated 28,268 transcripts for 6,380 RefSeq genes as well as 849 previously un-annotated loci. They investigated co-occurrence of alternative transcript features and, based on this analysis, report quite some interesting observations.

The presented set of full-length transcripts will be useful resource for improving annotation of rat genome, as a test set in evaluation of full length transcripts reconstruction and isoform quantification using short-read sequencing.

I read the manuscript with great interest and did not find any major drawbacks. I recommend it for publication in its current form.

Thank you for the positive comments and the recommendation.

Minor point:

Line 81: coding sequencing -> coding sequences

This typo has been corrected in the revised manuscript.

Reviewer #3 (Remarks to the Author):

This manuscript describes a hybrid workflow for reconstructing the rat hippocampal transcriptome, by taking advantage of the ability to sequence full-length transcripts offered by PacBio, while rectifying its relatively high error rate by correcting the sequences of its long reads through local alignments with short low-error Illumina sequencing reads. Moreover, comparison with independent 5' and 3'-end profiling allowed for the definition of reliable full-length transcripts, while integration with polysome/ribosome-profiling data enabled the prediction of actively translating isoforms. This hybrid workflow resulted in an increase in the number and accuracy of transcripts annotated in the rat genome, providing an important contribution to the characterization of the rat hippocampus transcriptomic and proteomic diversity. The manuscript's Introduction is also a very useful concise and up-to-date summary on the recent advances in sequencing technologies for studying the transcriptome, including the limitations of "third generation sequencing".

Overall, the article is very well written, the results of the work described therein are of great interest to the scientific community and we have no major concerns about its soundness.

Thanks for the positive comments.

We do, however, request that the following issues are addressed by the authors for the sake of improving clarity and preventing mis- and over-interpretations:

1. [Introduction, page 4, line 88] The concept of "coordinated alternative splicing" is not introduced and may not be familiar to the broad readership of Nature Communications. Given its biological relevance and the usefulness of the described methodologies in its study, coordinated alternative splicing should be explicitly (even if very briefly) defined.

We understand the reviewer's concern regarding the concept of "coordinated alternative splicing" because it may have a variety of meanings in different contexts. Following the reviewer's suggestion, we replaced "coordinated alternative splicing" to the more explicit term "co-occurring alternative splicing events" in the revised manuscript to avoid misunderstandings (p 4).

2. [Results, page 7, 1st paragraph, lines 140-141, and Figure 1a] Panel 1a does not visually discriminate the stated “experimental and computational arms”. The common reader will need to resort to the figure’s caption to learn this distinction. Accordingly coloring the flow chart’s rectangles would greatly facilitate the intended distinction.

We have colored the flow chart’s rectangles as suggested.

3. [Results, page 7, 2nd paragraph, lines 147-154] The authors unrealistically assume specialized knowledge from readers. Why were libraries size-fractionated and how are cDNA sizes expected to impact the accuracy of PacBio sequencing? Moreover, the concept of “subread” is not introduced – this can perhaps be done in the Introduction when talking about CCSs. Finally, “in general, the subread length in the four libraries” showing “a tendency to match the expected size” is an overstatement. For larger sizes (3rd and 4th panels of Supplementary Fig. 1a) it is more than “a fraction of reads” “outside of expected size ranges”.

Thanks for the suggestions. We have added the reasons for cDNA size fractionation in PacBio sequencing (p 7) and the concept of ‘subreads’ (p 5) in the revised manuscript, and have also rewritten the corresponding parts in the Results section (p 7).

4. [Results, page 7, 3rd paragraph, lines 156-164] The comparability between PacBio reads-per-gene and Illumina’s RPKMs is not entirely clear. Would plotting one against the other for the same genes result in something similar to Figure 1b? Given that RPKMs involve correction for transcript length (fewer reads for shorter transcripts, given the same expression level), does this variable introduce a bias contributing to normalization discrepancies?

To demonstrate the efficacy of the cDNA normalization step we introduced in the PacBio library preparation, in the original manuscript, we have (i) confirmed that the dynamic range of PacBio sequencing was indeed smaller than that of Illumina sequencing (Supplementary Figure 1c,d); and (ii) plotted a smoothed curve for the ratio between gene expression level (measured by Illumina FPKM) and number of PacBio reads per gene, against gene expression levels. However, we agree with the reviewer that the value on the y-axis was hard to interpret. To make it clearer, we have now plotted the ratio of PacBio read counts to gene expression level (estimated based on Illumina sequencing data) on the y-axis, instead. In addition to Figure 1b, where the x-axis represents the gene expression levels, we provided one more plot where the x-axis represents gene expression ranks (Supplementary Fig. 1e). Both plots clearly showed that we obtained relatively lower presentation of PacBio reads from highly expressed genes after cDNA normalization. This would avoid repetitively sequencing the same highly-expressed transcripts. As a result, introducing the normalization step could make PacBio sequencing cover more transcripts, and therefore more cost-efficient.

Following the reviewer’s suggestion, we also plotted the PacBio read counts against Illumina FPKM values for each gene (Fig R2). Although we observe a positive correlation between the two values, the fitted linear slope (blue dashed line) is smaller than the 1:1 increasing

slope (black dotted line). In particular, starting from genes with FPKM > 100, the PacBio read count stopped increasing. This indicates that the cDNA normalization in general worked, and the efficacy was the highest for the most highly expressed genes.

Figure R2. Scatter plot shows PacBio read counts against Illumina FPKM values for each gene. The blue dashed line indicates the fitted line of the data, and the black dotted line shows a diagonal line of slope 1.

Finally, using FPKM values in Illumina data is an attempt to address the length bias introduced in Illumina sequencing. In Illumina library preparation, there is a step to fragment RNA molecules into small pieces and each piece can potentially generate a certain number of reads in the sequencing data. Thus, longer transcripts tend to have proportionally more reads. In contrast, the PacBio library preparation did not include the RNA fragmentation step. Long transcripts and short transcripts have an equal probability to generate an identical amount of PacBio reads, hence the length normalization should not be applied.

5. [Results, page 9, lines 189-191, and Figure 2a] For clarity in the interpretation of the figure, it should be made explicit that the absence of a CAGE “ground truth” 5’ end explains the invalidated transcript.

We have modified the figure (Fig 2a) in the revision, where the reasons for invalidated transcripts are explicitly indicated. In addition to the preexisting invalidated example, we have included a parallel example, showing that invalidated transcripts could also result from the absence of 3’-seq cluster support. We have edited the figure legend accordingly as well.

6. [Results, page 11, lines 261-263, and Figure 3d and Supplementary Fig. 3c] The observed significant depletion of co-occurrence of intron retention and exon skipping should be revisited distinguishing alternative exons that fall within or neighbor introns with evidence for retention and those that do not, to control for the necessary mutual exclusivity of the two types of events when the associated exon and intron are adjacent. Also, given the reported increase in intron retention in non-neuronal genes during neuronal development (e.g. PMID 25258385), it would be very insightful to discriminate between introns in neuronal and non-

neuronal genes and check if the co-occurrence of multiple intron retention events is different between these two groups.

To address the first concern about the depletion of co-occurrence of intron retention and exon skipping, which might be entangled by alternative exons falling into neighboring introns, we counted the cases of co-occurrence of exon skipping and intron retention that were adjacent (a prototype shown in Fig. R3A) and the cases of mutually exclusive occurrence (prototypes shown in Fig. R3B). In total, we found 15 cases of co-occurrence and 31 cases of mutually exclusive occurrence. This agrees with the overall observation on the depletion of co-occurrence of intron retention and exon skipping. Moreover, the 15 cases of co-occurrence just take up 5% of the overall cases (297), so excluding adjacent exons and introns from the analysis does have any effect (Fig R3C).

Figure R3. Excluding co-occurrence (A) and mutual exclusivity (B) of exon skipping and intron retention in adjacent exons and introns, the co-occurrence map of alternative RNA processing events remains the same (C).

Regarding the second comment, we sorted the FLT transcripts into neuronal ($n=7203$), glial (7549), and others (6407) according to our 3'-end-seq data⁵. In all the three groups, approximately 10%~11% isoforms contain retained introns, suggesting that in the adult rat hippocampus there is no difference between neuronal genes and non-neuronal genes on the percentage of retained intron-containing isoforms. This observation differs from that in the suggested publication⁶, which is likely due to the fact that the adult rat hippocampus may have reached a steady state, and the intron retention-mediated gene regulation is not any more needed.

7. [Results, page 13, lines 297-300] This sentence suggests that exon skipping, intron retention, alternative 3' and 5' splice sites and alternative last exons are generally ORF-disrupting, which is not the case (e.g. most alternative exons keep the frame, with their length being a multiple of 3nt). Rephrasing is recommended in order to avoid imprecise interpretations of the conveyed message.

Following the reviewer's suggestion, we have rephrased the sentence to "these events could likely introduce frameshifts and thus disrupt the canonical open reading frames" in the revised manuscript (p 13). However, we could not agree to "most alternative exons keep the frame", as less than half of the alternative exons were of 3-divisible length in our data (Supplementary Fig. 2e).

8. [Results, page 14, lines 332-338] Rephrasing of this paragraph is also recommended. For instance, it is not clear what "appropriately coordinated RNA processing" means in that context. Deeming as "real" the isoforms observed in the active translation pool is somewhat misleading, as it suggests a ground truth and that others are "not real". Similarly, talking about "properly programmed RNA processing" suggests that alternatively processed isoforms are somewhat "not proper"... For instance, how does programmed RNA processing associate with ORF length? Is the "properly" programmed RNA processing isoform necessarily that producing the longer ORF?

As suggested by the reviewer, we have rephrased this paragraph to avoid misunderstanding in the revised manuscript (pp 14-15).

9. [Discussion, page 20, 1st paragraph, lines 460-463] It is not clear why the dependence of tools on gene annotation is "increasing".

This is an observation along the evolution of the software tools for RNA-seq data analysis. One of the underlying reasons for these tools tending to rely on known gene annotations is the computational challenges in transcriptome assembly imposed by short-read sequencing. For example, Steijger et al⁷ assessed dozens of transcript reconstruction tools, and found out that nearly all the tools had very low (<50%) transcript-level sensitivity and precision in assembling complex transcriptomes. However, we also found the sentence in our original manuscript was not relevant, so in the revised manuscript we have deleted it.

10. [Discussion, page 21, lines 484-487] Where are the data for the comparison of "the percentage of genes covered by our FLT transcripts across different lengths"?

We have now added a supplementary figure in the revised manuscript (Supplementary Figure 9).

11. [Discussion, page 21, lines 506-508] Stating that "co-occurrence of alternative donor and acceptor splicing sites indicates that the choice of upstream splice donor sites could influence

the selection of downstream splice acceptor sites” is almost a truism and not very insightful. Mentioning at least the hypothesis that alternative donor and acceptor sites could be regulated by the same trans-acting splicing factors would provide more food for thought for the splicing community.

Thanks for the suggestion. We have mentioned this hypothesis in the revised manuscript (p 21).

12. [Discussion, page 22, lines 508-510] Stating “that the defect in splicing out one intron may apply again to other introns in the same gene” implies that splicing efficiency tends to be gene-specific rather than intron-specific. This is a bold hypothesis that needs to be elaborated on. Moreover, taking intron retention as a splicing “defect” overlooks its regulatory role (e.g. PMID 25258385). Rephrasing and/or referencing is therefore advised.

Thanks for the suggestion. We have modified the corresponding discussion in the revised manuscript (p 22).

13. Figure 1d would greatly benefit for having the red/grey color code in a legend inside the panel and not only in the caption.

Thanks for the suggestion, which we have implemented in the revised manuscript.

We also suggest the following minor “proofreading” edits:

14. Figure 1c: there is a mismatch between the colors in the legend and the colors in the histograms themselves that make their interpretation quite tricky. The authors should look for a clearer and consistent way of discriminating the two distributions.

Thanks for pointing out this inconsistency, which was caused by transparent color settings in the histogram. In the revised manuscript, we have now switched to grey (before error correction) and red (after correction) colors, and also have added one sentence in the figure legend to clarify the color overlay.

15. The percentages of RefSeq-overlapping isoforms in Figure 3B add up to 101% and those of multiple alternative events to 62%, and not the 61% mentioned in line 249 (page 11). This is clearly just a “rounding” issue that should be fixed to prevent confusion.

Thanks for carefully reading the manuscript. We have now kept one more digit in the numbers in Figure 3b to avoid this rounding issue.

16. “Subread” is used in line 147 and “sub-read(s)” is used in line 617 and Figure 1c.

All corrected to ‘subread(s)’.

17. PacBio “reads-per-gene” is used in line 158 and “reads per gene” in line 1202.

Corrected to “reads per gene”.

18. Line 301, page 13: “significantLY” instead of “significant”.

Corrected.

Reviewer #4 (Remarks to the Author):

1. In the MS data analysis, the authors used a database derived from the FLT-based ORFoeme and a database based on RefSeq annotated ORFs for database searching. It looks like a combined database was used for the search, and the FDR estimation was performed without distinguishing novel and annotated peptides. It is well established that such analysis will underestimate the true FDR for novel peptides. The authors will need to perform separate FDR estimation to ensure the quality of the identified novel peptides.

We acknowledge the issue raised by the reviewer and addressed it by re-running two separate database searches.

Figure R4. Overlap of unique peptide sequences matched using the new (split FDR, 0.01 in yellow and 0.05 in green) and old (combined FDR, 0.01 in purple) database search.

As shown in the Venn diagram (Figure R4), the number of identified FLT peptides that pass at 1% split FDR went down by 382 (14.9%), and if the FDR for the new database search was set to 5%, we were losing only 209 (8.2%) peptides. We have added the result of the new database search at the split FDR 1% in the Supplementary Figure 7a.

2. To further illustrate the quality of the novel peptide identifications, the authors should compare the score distribution for novel peptides against that for annotated peptides. Either MS Amanda score or Percolator score can be used for the analysis.

We have plotted the MS Amanda peptide scores of the two distinct database searches for RefSeq and subsequently for the FLT database. Peptides matched to RefSeq are indicated in green and peptides matched to the FLT database are indicated in red. Figure R5 shows similar distributions of these scores for both database search peptide matches.

Figure R5: MS Amanda peptide score histogram for RefSeq (green) and FLT (red) database peptide search matches.

3. Was SILAC related modifications considered in database searching?

No SILAC modifications were used as no data from SILAC-labeled samples were used here.

4. Please report the numbers of protein sequences in the RefSeq and novel databases in the method section.

We have added the number of protein sequences in RefSeq and in our FLT in the Methods section (p 33).

5. How many annotated and novel peptides were identified under 1% peptide level FDR?

In the original manuscript, we have reported hits with 1% peptide level FDR, which matched 56233 and 2557 peptides in the RefSeq and FLT databases, respectively. For the new split FDR approach, we identified 53684 and 2175 peptides in the RefSeq and FLT databases, respectively. We have now added this information for clarity in the revised manuscript (Methods; p 34).

6. Please provide a table for all the identified novel peptides as a supplementary table, including spectrum title, m/z, charge, score, q-value, precursor delta mass, peptide sequence, modification and protein ID.

We have added a supplementary table (Supplementary Table 4) for all the identified novel peptides (both split- and combined-FDR searches) with full details.

7. Please provide annotated spectra for the peptides presented in Figure 6.

We have added Supplementary Figure 8 with all annotated spectra for the peptides presented in Figure 6.

8. Why the analysis only used “untreated/t0” data files from dataset PXD008596 instead of data from all time points?

In this study, we have already used a comprehensive data set (t0) to analyze the FLT peptides. The proteomics dataset comprised 518299 MS/MS spectra from 9 separate input files, matching 317809 or 308634 spectra for split- or combined-FDR database searches. Using SILAC data sets (i.e. data from other time points) would have required including a SILAC modification, dramatically increasing search space and altering the FDR. We thus did not make use of any SILAC data in this study.

9. I am confused about the active translating isoform prediction section. First, in the main text, it mentioned that “out of the 28,268 isoforms in our FLT dataset, 3,712 were excluded” but in the method section, “3,819 were excluded”. Second, it looks like the 28,268 isoforms were used for model training based on Ensembl annotation, and then the model was used to classify the same set of isoforms. How many isoforms were annotated by Ensembl? When there is a disagreement between Ensembl and the model assignment, which one should be trusted?

We are sorry for the confusion. The number of isoforms that have been excluded for translational status prediction was 3,712. We have now corrected the number in the Methods (p 32).

We used SVM models to predict the translational status of isoforms in our FLT. As a supervised approach, SVM models were trained by Ensembl isoforms by taking their biotype labels as the known translational status labels (i.e. “protein_coding” as translationally active, and the others as inactive). There were 23406 active isoforms and 2107 inactive isoforms in Ensembl. Then the Ensembl trained models were used to classify isoforms in our FLT. To avoid confusion, we have added a short introduction to SVM and reordered the training and prediction procedures in the Methods (pp 31-32).

10. When the authors compared the isoform and ORF diversity of the FLTs to that of RefSeq and Ensembl genes, was the analysis based on the overlapping set of genes?

Yes, only the overlapping set of genes was considered, to make the comparison a fair one. We have added this important information to the revised manuscript (p 33).

11. In the analysis of co-occurrence of alternative events, RefSeq was used as the reference, which might be misleading. It is unclear whether each transcript sequence in RefSeq comes from an individual sample in a specific condition.

We aim to identify the simultaneous alternative events of each FLT isoform when compared to their reference isoforms. We chose the closest RefSeq isoforms for each FLT isoform as the reference because (i) the RefSeq isoforms were manually curated, forming a set of authentic isoforms existing in the organism; and (ii) in nearly all genes, there is only one RefSeq isoform annotated, which is by and large the most abundant one across multiple tissues. As pointed out by the reviewer, taking the closest RefSeq isoforms as the reference might under- or over-estimate the co-occurrence of alternative events, since (i) the annotated isoforms may not express in rat hippocampus, and (ii) the true reference isoforms expressed in the rat hippocampus may not be included in RefSeq annotation. To accommodate the reviewer's comment here, we performed additional analysis by taking the major isoform (i.e. the most abundant one) of each gene in our FLT collection as the reference. The result was similar to that by taking RefSeq as the reference, and has been added in the revised manuscript (Supplementary Figure 3f).

References

- 1 Au, K. F., Underwood, J. G., Lee, L. & Wong, W. H. Improving PacBio long read accuracy by short read alignment. *PLoS One* **7**, e46679, doi:10.1371/journal.pone.0046679 (2012).
- 2 Salmela, L. & Rivals, E. LoRDEC: accurate and efficient long read error correction. *Bioinformatics* **30**, 3506-3514, doi:10.1093/bioinformatics/btu538 (2014).
- 3 Fu, S., Wang, A. & Au, K. F. A comparative evaluation of hybrid error correction methods for error-prone long reads. *Genome Biol* **20**, 26, doi:10.1186/s13059-018-1605-z (2019).
- 4 Irimia, M. *et al.* A highly conserved program of neuronal microexons is misregulated in autistic brains. *Cell* **159**, 1511-1523, doi:10.1016/j.cell.2014.11.035 (2014).
- 5 Tushev, G. *et al.* Alternative 3' UTRs Modify the Localization, Regulatory Potential, Stability, and Plasticity of mRNAs in Neuronal Compartments. *Neuron*, doi:10.1016/j.neuron.2018.03.030 (2018).
- 6 Braunschweig, U. *et al.* Widespread intron retention in mammals functionally tunes transcriptomes. *Genome Res* **24**, 1774-1786, doi:10.1101/gr.177790.114 (2014).
- 7 Steijger, T. *et al.* Assessment of transcript reconstruction methods for RNA-seq. *Nat Methods* **10**, 1177-1184, doi:10.1038/nmeth.2714 (2013).

REVIEWERS' COMMENTS:

Reviewer #1 (Remarks to the Author):

The revisions have addressed my major concerns with the original manuscript.

Reviewer #3 (Remarks to the Author):

The authors have satisfactorily addressed all my concerns.

My only minor discretionary proof-editing suggestion is that apostrophes (') are replaced by actual primes (´) when 5-prime and 3-prime splice sites and UTRs are mentioned.

Apart from this, I am happy with the manuscript being published in its current form.

Reviewer #4 (Remarks to the Author):

The authors have adequately addressed my comments.

Point-by-point response to the reviewers' comments

Reviewer #1 (Remarks to the Author):

The revisions have addressed my major concerns with the original manuscript.

We appreciate the positive feedback from this reviewer.

Reviewer #3 (Remarks to the Author):

The authors have satisfactorily addressed all my concerns.

My only minor discretionary proof-editing suggestion is that apostrophes (') are replaced by actual primes (ˆ) when 5-prime and 3-prime splice sites and UTRs are mentioned.

Apart from this, I am happy with the manuscript being published in its current form.

We appreciate the positive feedback from this reviewer, and have changed all the prime (ˆ) notations throughout our manuscript.

Reviewer #4 (Remarks to the Author):

The authors have adequately addressed my comments.

We appreciate the positive feedback from this reviewer.